# Modelling the Effects of Changes in Forest Cover and Climate on Hydrology of Headwater Catchments in South-Central Chile

**Guillermo Barrientos [1],\*** , **Albert Herrero [2]**, **Andrés Iroumé [3]**, **Oscar Mardones [4]** and **Ramon J. Batalla [2,3,5]**

[1] Faculty of Forest Sciences and Natural Resources, Graduate School, Universidad Austral de Chile, Independencia 631, Valdivia 5110566, Chile

[2] Catalan Institute for Water Research, Girona, 101-17003 Catalonia, Spain; aherrero@sicenginyeria.com (A.H.); rbatalla@macs.udl.cat (R.J.B.)

[3] Faculty of Forest Sciences and Natural Resources, Institute of Conservation, Biodiversity and Territory, Universidad Austral de Chile, Independencia 631, Valdivia 5110566, Chile; airoume@uach.cl

[4] Forestal Mininco S.A., Los Ángeles 4450808, Chile; oscar.mardones@forestal.cmpc.cl

[5] Fluvial Dynamics Research Group (RIUS), University of Lleida, Lleida, 1-25003 Catalonia, Spain

\* Correspondence: guillermo.barrientos@alumnos.uach.cl; Tel.: +56-63-229-3004

**Abstract:** This study analyses the changes in the runoff of forested experimental catchments in south-central Chile, to determine to what extent observed trends can be attributed to effects of intensive forestry and/or climate change. For this, we applied the distributed TETIS® model to eight catchments (7.1−413.6 ha) representative of the land uses and forestry activities in this geographical area. Rainfall and runoff data collected between 2008 and 2015 were used for modelling calibration and validation. Simulation of three land uses (current cover, partial harvest and native forest) and 25 combinations of climatic scenarios (percentage increases or decreases of up to 20% of rainfall and evapotranspiration relative to the no-change scenario applied to input series) were used in each calibration. We found that changes in land use and climate had contrasting effects on runoff. Smaller catchments affected by the driest climatic scenarios experienced higher runoff when the forest cover was lower than under full forest cover (plantations or native forests). In contrast, larger catchments under all climatic scenarios yielded higher runoff below the full forest cover than under partial harvest and native forest. This suggests that runoff can be influenced, to a great extent, by rainfall decrease and evapotranspiration increase, with the model predicting up to a 60% decrease in runoff yield for the dry's climatic scenario. This study proves to be relevant to inform ongoing discussions related to forest management in Chile, and is intended to minimize the impact of forest cover on runoff yield under uncertain climatic scenarios.

**Keywords:** TETIS® model; land use scenarios; climatic scenarios; runoff; forestry; Chile

---

## 1. Introduction

Changes in land use affect several components of river flow regimes, including base flows, mean discharge and flood magnitude and frequency. Changes in land management may result in altered runoff patterns, which in turn can be intensified by changes in climate [1]. Understanding the role of soil cover on trends in the catchment hydrological response has been of great interest in recent decades. Direct rainfall runoff models e.g., [2], distributed and generalized hydrological models e.g., [3,4], and physically-based models e.g., [5], provide insights on the impacts of land cover and climate fluctuations on runoff. The interest in using conceptual models may be attributed to a significant

increase in forest management [6]. Forest expansion and climate change have been identified as two of the main factors driving the reduction in runoff at local and global scales [7], modifying processes such as infiltration, rainfall interception, and evapotranspiration [8], and accounting for losses in the runoff yield, and reduction in average flow and annual runoff storage contribution. The mid-latitude terrestrial ecosystems of the western coast of Chile are characterized by a climate with dry summers occupied by highly productive forests of fast-growing exotic species (*Eucalyptus nitens* and *Pinus radiata*) which currently occupy 2.4 million hectares e.g., [9–11]. Runoff resources are controlled by a marked interannual and seasonal variability in rainfall, whereas climate change has resulted in an increase in temperatures and decrease in rainfall [12].

While the effects of intensive forest management and climate change on runoff seem clear in the northern hemisphere [13,14], the interpretation of the results on southern mid-latitude ecosystems such as those in Chile is far from concluded, particularly when comparing different forest management systems. Globally, it is accepted that, in many cases, the expansion of forest plantations on former agricultural land results in a decreased streamflow, especially during the dry season [15,16], reducing not only runoff yield but also groundwater storage [17,18]. In addition, a decrease in the annual runoff yield, summer runoff and peak flows is associated with higher evapotranspiration from the forested portion of the catchments, as previously reported elsewhere, e.g., mostly Bosch and Hewlett [19], Best et al. [20], Andréassian [21], Brown et al. [22]. From the ecological and socio-economic points of view, the expansion of planted forests has been questioned because of its impact on decreased streamflow, especially during the dry season [23]. Understanding the relationship between catchment runoff, forest management, and climatic factors, as well as their temporal evolution, is therefore crucial in developing integrated water management policies in forested catchments, especially in areas where water shortages are structural and frequent.

From the forest management perspective in Chile, the study of these type of catchments can inform on future responses to further changes in climate and land use. The aim of this manuscript is to determine the relationship between catchment runoff and the factors of forest management and climatic change, by simulating a variety of combinations of these factors. For this, we applied the TETIS® hydrological model to eight catchments that are representative of the region. Rainfall and runoff data collected in these catchments between 2008 and 2015 were used for model calibration and validation. Two different calibrations were performed: in the first, the 2009–2012 discharge series was considered for the computation of the Nash–Sutcliffe index (NSE), whereas in the second, data from the wet season were excluded. This approach was adopted in order to improve the simulation of both the global time series and the results of the dry season specifically. The hydrologic model was also used to simulate the effects of potential land use and management changes and climatic scenarios. Three land uses and twenty-five climate scenarios were implemented for each calibration, for a total of 1200 simulations. This large number of simulations allows for a comparison of the hydrological response for forest catchments with different characteristics, to present change projections and evaluate the impacts of global change on the annual distribution of rainfall runoff, with special attention to summer runoff, daily runoff, runoff yield, and their respective runoff coefficients.

## 2. Materials and Methods

### 2.1. Study Area

The study was conducted in eight experimental catchments in the Chilean Coastal Range, in the Biobío region, all of them monitored between 2008 and 2015 (Figure 1), 3 km west of the town of Nacimiento (37°28' S, 72°42' W) [24]. The study catchments are labelled N2, N3, N4, N5, N7, N8, N9 and N11 (see Figure 1 for location details); the catchment numbers were previously assigned as part of a larger experimental network comprising 15 catchments in the area [24].

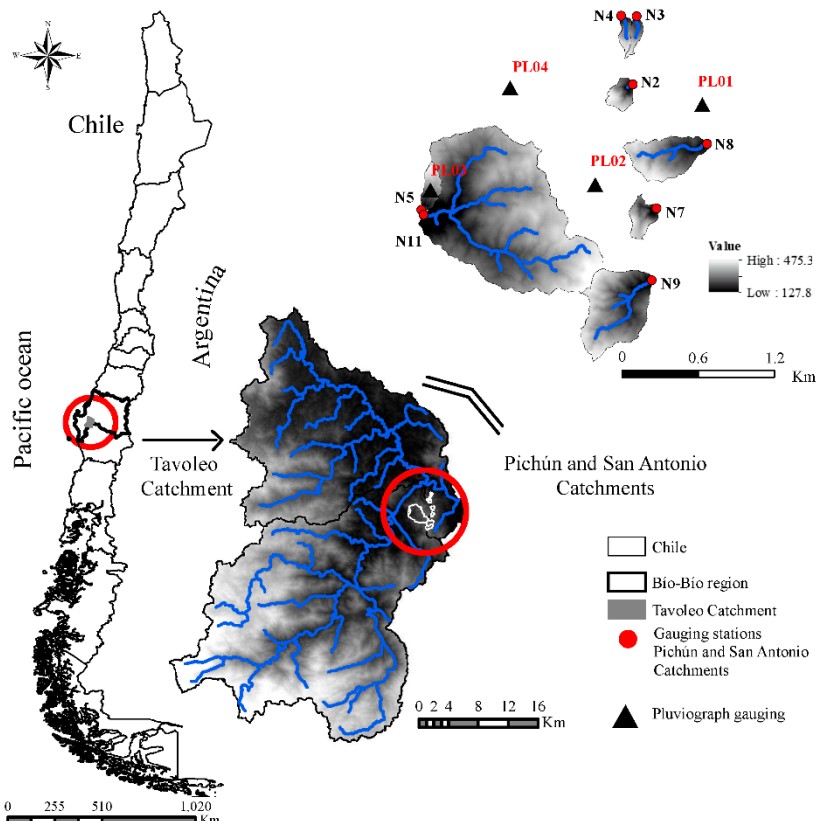

**Figure 1.** Experimental catchments located in the Chilean Coastal Range (37°28′ S, 72°42′ W), Biobío region.

Catchment areas vary between 7 and 414 ha with altitudes ranging between 127 and 474 m a.s.l. The average catchment slope is between 36 and 44% (the median slope value for all grid cells in a catchment) [24]. The region is characterized by a Mediterranean climate with dry summers [24]. Mean annual rainfall is 1160 mm (ranging from 866–1421 mm; data from 2008 to 2015), most of which (95%) occurs between April and September during frequent and prolonged low- to moderate-intensity frontal storms. The long-term rainfall record is marked by inter-annual variations and its spatial distribution is orographically controlled by the topography of the mountain range. During rainfall events, the plantations are generally immersed in mist or clouds due to the relatively low altitude of the catchments (233–389 m.a.s.l) [24]. Climate has wet-mild winters, and exceptionally dry and warm summers. The temperature ranges from more than 40 °C during the summer to less than −3 °C in winter with an annual average of 13 °C [25]. The mean annual runoff coefficient in the catchments for the 2008–2015 period is 25.5% (between 16.3 and 41.3%) [24].

The geology of the area is composed by a diorite formation (i.e., granitoids from the Cretaceous), which has experienced intense metamorphism. The predominant soil type in the area is Luvisol with low clay content in surface soil; with variable structure due to fragments of bedrock within the topsoil and higher clay content in the subsoil, with high saturation and depth of approximately 160 cm [24,25]. The total depth of the unconsolidated and fractured material exceeds 700 cm [25] and the saprolite-clayey formation reaches depths of 560 ± 215 cm [24,26]. Field studies confirm that the saprolite is highly permeable, and percolation has been observed on road cuts even during dry summer months. The vegetation cover varies from 56.3 to 92.7% for planted forests and from 4.0 to 40.9% for natural forests. Most of the catchment areas correspond to a forest cover covered by *Eucalyptus*. The steep hillslope sections are rather short and occupied by *Pinus radiata* specimens that have invaded the riparian zone by natural regeneration during previous rotations [27]. The soil exhibits a thin herbaceous cover (<25%) mainly composed of grasses that usually perish during the summer because

of water deficit [10]. The shrub cover under the most developed plantations is rather dispersed and mostly composed of the genera *Aristotelia* and *Rubus*, as well as some arboreal genera such as *Luma*, *Peumus*, *Persea lingue* and *Nothofagus*. The fluvial channel in all catchments is bounded by a strip of forest with an average width between 15 and 70 m and composed of native forest species of the genera *Luma*, *Peumus*, *Persea lingue* and *Nothofagus*. The remaining catchment area is dominated by plantation forests of the genera *Eucalyptus* and *Pinus* (Table 1) [24].

**Table 1.** Physiographic and land use characteristics of the eight studied catchments (see Figure 1 for location details).

| Parameter | N2 | N3 | N4 | N5 | N7 | N8 | N9 | N11 |
|---|---|---|---|---|---|---|---|---|
| Species | *Pinus radiata* | *Eucalyptus* spp. | *Eucalyptus* spp. | *Eucalyptus* spp. | *Eucalyptus* spp. | *Eucalyptus* spp. | *Eucalyptus* spp. | *Eucalyptus* spp. |
| Type of soil | Clayey to loamy | Clayey to loamy | Clayey to loamy | Clayey to loamy | Clayey to loamy | Clayey to loamy | Clayey to loamy | Clayey to loamy |
| Lithology | Granite-basalt | Granite-basalt | Granite-basalt | Granite-basalt | Quartzite-schist | Granite-basalt | Quartzite-schist | Granite-basalt |
| P (mm) (range) | | | | 866.7–1421 | | | | |
| Min-Max Temperature (°C) | 1.2–37.2 | 3.3–36.2 | 2.5–35.3 | 2.8–40.1 | 1.6–32.1 | 0.7–33.3 | 3.3–36.1 | 0.7–38.3 |
| dVmax [1] | 792.4 | 752.8 | 810.2 | 840 | 834.7 | 1016 | 666.6 | 768.3 |
| DAF [2] | 0.57 | 1.54 | 1.39 | 0.52 | 0.41 | 0.89 | 1.36 | 0.74 |
| DPF [2] | 8.1 | 34.5 | 19.9 | 8.3 | 3.3 | 7.6 | 9.7 | 9.5 |
| DMF [2] | 0.02 | 0.12 | 0.02 | 0.09 | 0.1 | 0.09 | 0.44 | 0.08 |
| Catchment area (ha) | 13.9 | 7.1 | 7.6 | 14.2 | 16.9 | 54.9 | 98.3 | 413.6 |
| Catchment slope (m/m) | 0.27 | 0.4 | 0.42 | 0.44 | 0.29 | 0.36 | 0.39 | 0.38 |
| Drainage density (km/km$^2$) | 2.4 | 5.8 | 5.2 | 2.4 | 3.1 | 2.8 | 2.6 | 2.2 |
| Mean elevation (m a.s.l) | 323 | 233 | 234 | 248 | 360 | 269 | 368 | 300 |
| Topographic relief (m) | 86 | 173 | 169 | 236 | 93 | 195 | 214 | 347 |
| Channel gradient (m/m) | 0.08 | 0.08 | 0.09 | 0.08 | 0.27 | 0.14 | 0.18 | 0.27 |
| sL/sG [3] | 0.31 | 0.21 | 0.22 | 0.18 | 0.92 | 0.34 | 0.44 | 0.71 |
| Roughness | 0.21 | 1.00 | 0.88 | 0.57 | 0.29 | 0.55 | 0.56 | 0.76 |
| Percent Roads % | 3.2 | 1.7 | 2.0 | 1.8 | 1.5 | 1.8 | 2.6 | 3.0 |
| Percent Plantation % | 92.7 | 75.4 | 86.8 | 77.2 | 86.5 | 83.4 | 56.3 | 65.8 |
| Percent Natural forest % | 4.0 | 22.9 | 10.4 | 20.6 | 9.7 | 14.5 | 40.9 | 19.9 |
| Grassland % | 0.0 | 0.0 | 0.9 | 0.4 | 0.0 | 0.3 | 0.0 | 1.0 |
| Percent Harvested % | 0.0 | 0.0 | 0.0 | 0.0 | 2.3 | 0.0 | 0.1 | 10.2 |
| Biomass volume (m$^3$/ha) | 295.1 | 302.6 | 320.9 | 152.3 | 179.1 | 150.5 | 170.7 | 163.6 |
| Plantation density (No. tree/ha) | 315 | 369 | 342 | 1160 | 1174 | 567 | 1320 | 1246 |
| Age plantation | 28 | 5 | 5 | 8 | 3 | 3 | 15 | 9 |
| Width riparian zone (m) | 15.5 | 35.7 | 17.7 | 40.4 | 23.9 | 21 | 49.7 | 70.5 |

[1] dVmax: annual dynamic storage [24]. [2] Average annual daily flows (ADF), daily peak flow (DPF), daily minimum flow (DMF) [24]. [3] Average slope of channel/average slope of the catchment (sL/Sg).

**Table 2.** Time series of rainfall, runoff and temperature, correspond to the period 1 April 2008 and 31 March 2015.

| * Hydrological Year | Annual Precipitation (mm) | Daily Maximum Rainfall (mm) | Average Temperature (°C) | Annual Evapotranspiration (mm) | Annual Runoff (mm/a) | | | | | | | |
|---|---|---|---|---|---|---|---|---|---|---|---|---|
| | | | | | N2 | N3 | N4 | N5 | N7 | N8 | N9 | N11 |
| 2008–2009 | 1420 ± 13.9 | 146 | 14.5 ± 5.6 | 1065 ± 2.2 | 609 | 555 | 504 | 207 | 172 | 324 | 195 | 744 |
| 2009–2010 | 1421 ± 10.7 | 81 | 13.5 ± 5.0 | 966 ± 1.8 | 351 | 424 | 357 | 675 | 152 | 283 | 574 | 519 |
| 2010–2011 | 867 ± 6.9 | 73 | 13.8 ± 4.9 | 1015 ± 1.8 | 248 | 423 | 489 | 391 | 140 | 252 | 469 | 385 |
| 2011–2012 | 1140 ± 8.2 | 51 | 14.2 ± 5.5 | 1070 ± 2.1 | 213 | 367 | 380 | 230 | 145 | 212 | 297 | 214 |
| 2012–2013 | 1085 ± 10.0 | 112 | 14.4 ± 4.9 | 1133 ± 2.0 | 162 | 268 | 178 | 167 | 147 | 185 | 279 | 161 |
| 2013–2014 | 955 ± 8.5 | 76 | 13.6 ± 5.4 | 1234 ± 2.2 | 114 | 183 | 118 | 105 | 133 | 203 | 361 | 137 |
| 2014–2015 | 1255 ± 0.2 | 82 | 11.5 ± 6.7 | 1082 ± 1.9 | 211 | 307 | 245 | 192 | 297 | 350 | 497 | 285 |

± for reference standard deviation is also show * Hydrological year: Considering Hydrologic runoff Year, 1 April to 31 March.

*2.2. Field Data*

Time series of temperature, rainfall and runoff were directly obtained in the field, recorded with a 6-minute resolution and summarized as daily values [24] (Table 2) between 1 April 2008 and 31 March 2015. As this study is focused on the seasonal and yearly scales, a daily resolution for analysis was adopted, which required reasonable computational times. Temperature series were not directly used by the model (no snow coverage was considered) and were only considered to calculate evapotranspiration (i.e., calculated by means of the equation by Hargreaves and Samani [28] and Barrientos and Iroumé [24].

*2.3. Hydrological Modelling*

2.3.1. TETIS® Model

The hydrological TETIS® v9.0 model was used to analyse the runoff yield in each of the catchments under different climate and land-use scenarios. TETIS® is a distributed model developed by the Technical University of Valencia, Spain e.g., [29] which, among others, includes hydrological, sedimentological and nutrient modules (the last two of which are beyond the scope of this study). The hydrological module is based on a grid structure where input data are provided in a raster format describing the spatial distribution of the different variables. Each cell is conceptualized as five connected tanks, which are used to establish a water mass balance related to the different hydrological processes: rainfall, vegetation interception, soil water storage, soil infiltration, direct runoff generation, interflow, groundwater storage, baseflow and water losses (by evapotranspiration). Mass balance equations are used to reproduce these hydrological processes, including parameters related to different physical properties (i.e., maximum canopy interception, soil water holding capacity, infiltration capacity). Cells are classified as one of three different types: hillslope, gully and river channel. The geomorphologic kinematic wave method [29,30] was used to propagate the hydrograph along the channel network. Nine geomorphologic parameters of the catchments were used in the equations for the accumulated area, discharge, and effective width e.g., [31,32]. This model has been used satisfactorily for different research and management purposes such as impact of climate change on, runoff trend and sediment yield e.g., [6,33–36].

2.3.2. Input Data

The temperature was measured by one weather station and rainfall in four gauges. A database of precipitation and temperature was developed in a mesh of 1 × 1 m in ArcGIS®, was used Kriging interpolation, calculating the square difference between the values of the associated locations that extends throughout the catchments [37]. These data, together with the runoff measured at gauging stations located at the catchment outlets, were used as inputs for the hydrological model. Digital elevation models (DEM) were derived from LiDAR® data (at 1-m resolution) using the toolbox Topo to Raster (i.e., interpolation) within ArcGIS®. Three DEMs with cell resolutions of 1, 5 and 10 m were initially generated [24]. The 1 and 10 m resolutions presented error values during calibration (i.e., in raster information, columns and rows and statistics values, minimum and maximum on slope, flow accumulation and flow direction) and, consequently, the 5-m resolution was set for all the catchments.

The DEM was used to derive slope, flow direction, flow accumulation and flow surface velocity using ArcGIS®: (i) slope identified the steepest downhill descent from a given cell, i.e., the maximum change in elevation over the distance between the cell and its eight neighbours; (ii) flow direction was defined as the direction of the steepest descent, or the maximum fall from each cell; (iii) flow accumulation was the accumulated weight of all the cells flowing to each cell with descending slope in the output raster; and finally, iv) surface velocity was calculated using Equation (1):

$$V = 1.4 \times slope^{0.5} \tag{1}$$

Information related to land use and vegetation cover was derived from photointerpretation of (tVIS) LIDAR® images [24]. Land uses were defined as (i) forest plantation, (ii) native forest and (iii) harvested land, and digitized in polygons using the vector layer method; the respective areas were subsequently calculated in relation to the total area of each catchment, and the relative weight of each land use (in percentage) was obtained. The identification of planted tree species was performed by image recognition, supported by ground-truth validations; six 10 m wide and 20 m long transects located outside the riparian zone were used for validation on both sides of the streams per catchment. Finally, soil properties required by TETIS® as model input were derived as raster files (5-m resolution) from the Geological Map of Chile (scale 1:1,000,000) [38], and included vertical and horizontal permeability of the surface and sub-surface layers and vertical permeability of the deeper soil, to account for water losses into the aquifer [24].

### 2.3.3. Calibration and Validation

The TETIS® hydrological module is based on nine parameters included in the mass balance equations, and requires calibration. These parameters are introduced to minimize the uncertainty related to static storage, evapotranspiration, infiltration, overland flow, percolation, interflow, deep aquifer flow, connected aquifer flow and kinematic flow velocity. Data from 1 April 2008 to 31 March 2009 were used to obtain realistic initial conditions for the calibration period, defined as 1 April 2009 to 31 March 2012. Calibration was performed independently for each of the eight catchments and was based on the discharge data obtained from gauging stations. The goodness of fit between simulated and observed discharges ($Q_s$ and $Q_o$, respectively) was evaluated by means of the widely used NSE [39] (Table 3).

**Table 3.** Nash–Sutcliffe Index (NSE) value for Calibration and validation.

| | Catchments | | | | | | | |
|---|---|---|---|---|---|---|---|---|
| | N2 | N3 | N4 | N5 | N7 | N8 | N9 | N11 |
| NSE Calibration (*whole-time series* calibration) | 0.5 | 0.3 | 0.4 | 0.6 | 0.4 | 0.4 | 0.5 | 0.5 |
| NSE Validation (*whole-time series* calibration) | 0.3 | −2.8 | −1.9 | −2.8 | 0.1 | 0.5 | −1.8 | −1.2 |
| NSE Calibration (*dry season* calibration) | 0.6 | 0.6 | 0.2 | 0.5 | 0.4 | 0.4 | −0.2 | 0.5 |
| NSE Validation (*dry season* calibration) | 0.3 | −0.1 | −0.6 | 1.4 | −3.2 | −3.1 | −17.3 | −2.0 |

Two different calibrations were performed. First, the entire discharge series 1 April 2009 to 31 March 2012 was considered for the computation of NSE and the optimization of fit between the observed and simulated data (hereafter *whole-time series* calibration). Afterwards, a second calibration was performed, which excluded the wet April to September season data (hereafter *dry season* calibration). This approach was adopted to obtain reproduction of discharges during the dry season, as the assessment of water resources during this period is the main focus of this study due to its importance from ecological and socio-economic standpoints. After completing the calibration, the model was validated using the time series 1 April 2012 to 31 March 2015. The NSE was applied to test the quality of the simulated results, on a daily time step and end of each data series, using the following rating criteria [40]: not satisfactory (NSE ≤ 0.50), satisfactory (0.50 < NSE ≤ 0.70), good (0.70 < NSE ≤ 0.80) and very good (NSE > 0.80).

### 2.3.4. Simulated Scenarios

The hydrologic model calibrated using the *whole-time series* calibration and *dry season* calibration was used to simulate the effect on runoff for different land use and climate scenarios. Three land use and twenty-five climate scenarios were implemented for each calibration, which lead to a total of 1200 simulations.

(a)  Land use scenarios

Three land cover scenarios were considered (i.e., additional supporting information on the three land cover scenarios can be found online in the Supporting Information section at the end of the paper, Figure S1): (i) Current cover, represented by the land use map for the year 2015; (ii) Partial harvest, considering that 50% of plantation area is removed in each of the study catchments and remains bare; and (iii) Native forest, representing an increase of native forest cover replacing the plantation previously clear cut (as per scenario partial harvest).

(b)  Climate scenarios

A *scenario-neutral approach* was followed by applying uniform proportional increments of several magnitudes to the historical time series of rainfall (P0) and evapotranspiration (E0) (Additional supporting climate scenarios information may be found online in the Supporting Information section at the end of the article Table S1), similarly to previous studies [6,41,42]. Percentage increments of ±10% and ±20% were applied to both time series, leading to 25 combinations of rainfall−evapotranspiration scenarios. The chosen increases for rainfall and evapotranspiration are in the order of the expected increments for the 21st century RCP8.5 scenarios (2006−2099) in Chile [12]. This indicates decreases in the annual runoff of approximately 40% by the end of the century, higher than projected precipitation decreases (up to 30%) and temperature increases (2.6−3.2 °C) near the coast. Some scenarios considered in this study are not plausible (i.e., increment or a decrease of precipitation by 20% and increment or a decrease of evapotranspiration by 20%), but the simulations were performed to provide a framework for a sensitivity analysis of the hydrological response of catchments to climate change.

2.3.5. Data Grouping and Analysis

Furthermore, we set up four groups of catchments for the data analyses based on land cover and catchment size to assess the effects of different land cover and climatic scenarios on catchments of different sizes (i.e., additional supporting information on data grouping and analysis can be found online in the Supporting Information section at the end of the paper, Table S2).

We developed runoff duration curves (RDC) from daily data for each of the catchments to characterize the temporal distribution of the runoff and detect changes within the various simulations. For this, we used the observed streamflow from 1 April 2008 to 31 March 2015 and values from the modelling exercise under the different land use and climatic scenarios. From RDCs we obtained percentile values of $Q_5$, $Q_{16}$, $Q_{50}$, $Q_{84}$ and $Q_{95}$; where $Q_i$ is the value corresponding to ith percentile, i.e., the runoff equal to or exceeding 5, 16, 50, 84, and 95% of the time, respectively. In addition, we calculated the runoff standard deviation (RSD; Equation (2)) relative to the median daily runoff to analyse particular hydrological trends in the study period; this was adapted from the index developed by Folk and Ward [43] for fluvial sediments and further applied by Batalla et al. [44] to assess effects of reservoir operation on streamflow. Mann–Whitney (MW) and Kruskal–Wallis (KW) non-parametric tests were used to derive the statistical significance of differences ($p \leq 0.05$) of runoff yield values and RDC percentiles between groups and catchments, combining land use and climate scenarios.

$$RSD = \left( (Q_{84} - Q_{16}) + \frac{Q_{95} - Q_5}{Q_{50}} \right) \qquad (2)$$

Finally, we analysed the variation of annual rainfall, annual runoff, and annual runoff coefficients (i.e., the total runoff as a proportion of the total rainfall) using the MW (rank-sum test) for two independent catchments and the KW non-parametric test for more than two groups of catchments. Statistical significance was assigned at $p < 0.05$.

## 3. Results

### 3.1. Characterization Observed Data of Rainfall and Runoff

The annual rainfall was between 866 and 1421 mm across the study years and catchments, with daily maximum rainfall intensities between 51 mm and 146 mm (Table 2). Rainfall between April and September (winter season) accounted for 77% of total annual rainfall. Annual runoff varied between 105 and 1206 mm/y (KW test, $p = 0.02$), and the annual runoff coefficient ranged between 0.11 and 0.85 (KW test, $p = 0.01$) (i.e., additional supporting information on annual runoff and runoff coefficient can be found online in the Supporting Information section at the end of the paper, Table S3). In particular, the annual runoff was similar between catchments in N3; N4 (MW test, $p = 0.5$), N5; N7 (MW test, $p = 0.06$) and N8; N9; N11 (KW test, $p = 0.2$). The annual runoff of catchment N2 (230.5 mm/year), the only one in group 1, was not statistically different from that of catchments within the other groups (KW test, $p \leq 0.1$) (Figure 2a). The runoff coefficient was also statistically similar between catchments within group 2 (MW test, $p = 0.5$), group 3 (MW test, $p = 0.06$) and group 4 (KW test, $p = 0.16$).

Overall, the hydrograph at the outlet of the catchments exhibited a rapid response to rainfall, resulting in high runoff, especially during the wettest months. The shape of the flow duration curves of N5, N7, N8, N9 and N11 is almost identical (Figure 2b), with attenuated slopes from $Q_{16}$ to $Q_{84}$. Catchments N2, N3 and N4 showed more extended low flow ($>Q_{95}$) periods than the rest of the catchments. $Q_5$, $Q_{16}$, $Q_{50}$, $Q_{84}$ and $Q_{95}$ were statistically similar for catchments within group 2, group 3 and group 4 (MW test; 0.6 and 0.08, respectively; $p = 0.8$). In contrast, values of catchment N2 (group 1) were statistically different from those of group 4 (KW test, $p = 0.01$).

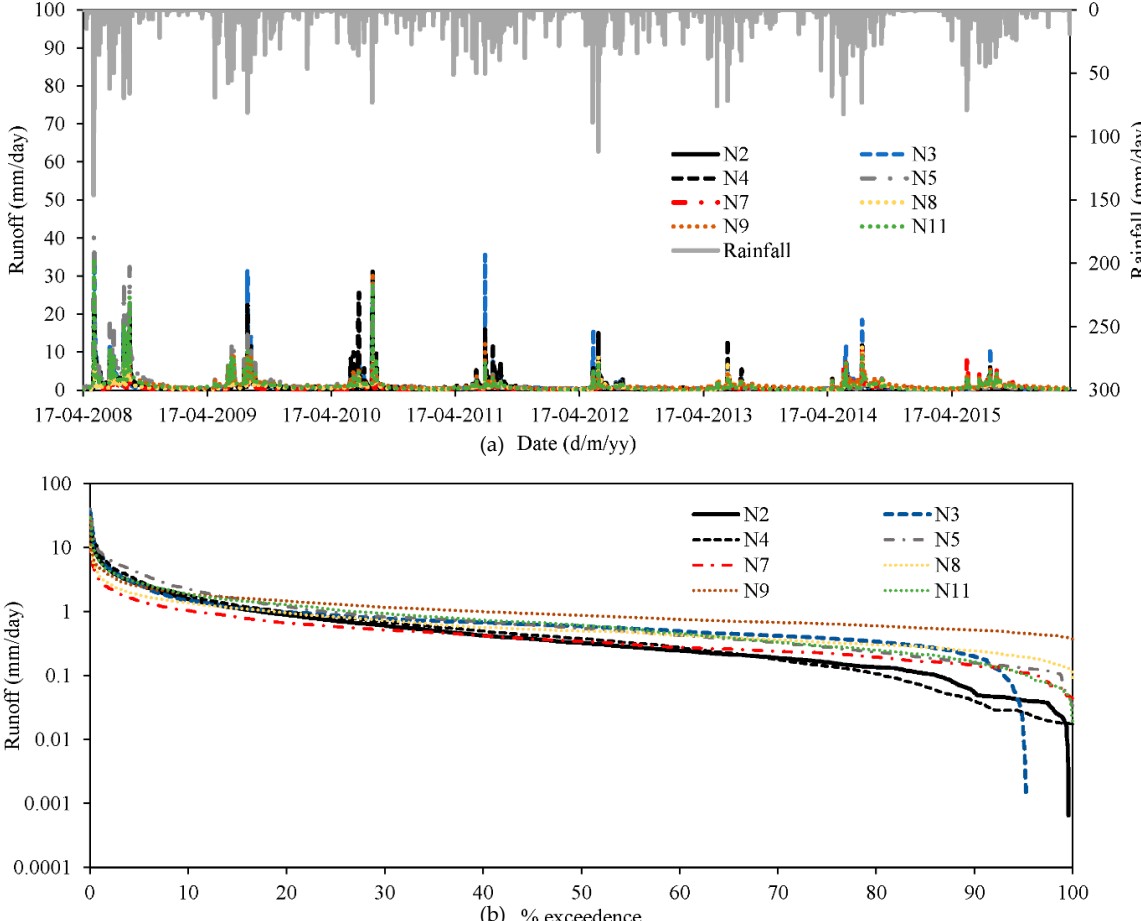

**Figure 2.** (**a**) Rainfall and runoff observed at the catchment's outlets. (**b**) Flow Duration Curves for each catchment (All curves were developed using observed daily data).

## 3.2. Model Performance

### 3.2.1. Whole-Time Series Calibration

Using the whole-time series calibration and land use for the year 2015, mean Qs of each of the four catchments groups (see Supporting Information Table S2) showed satisfactory agreement with Qo in N2, N5, N9 and N11 (Table 3). The hydrograph shape was overall correctly reproduced, despite divergences for higher flow events, as peak discharges were underestimated by the model (Figure 3), and for discharges during the dry season, which were also underestimated. Results for the validation period showed an unsatisfactory adjustment between $Q_s$ and $Q_o$, indicating that the hydrograph was correctly reproduced, but in this case, simulated peak runoffs were often higher than the observed runoffs, and some $Q_o$ peaks were not reproduced; moreover, the simulated runoffs during dry months were also underestimated. The NSE ranged between 0.3 and 0.6, with $NSE_{mean} = 0.5$ for the calibration period, and between −2.8 to 0.5, with $NSE_{mean} = −1.2$ for the validation period (Table 3).

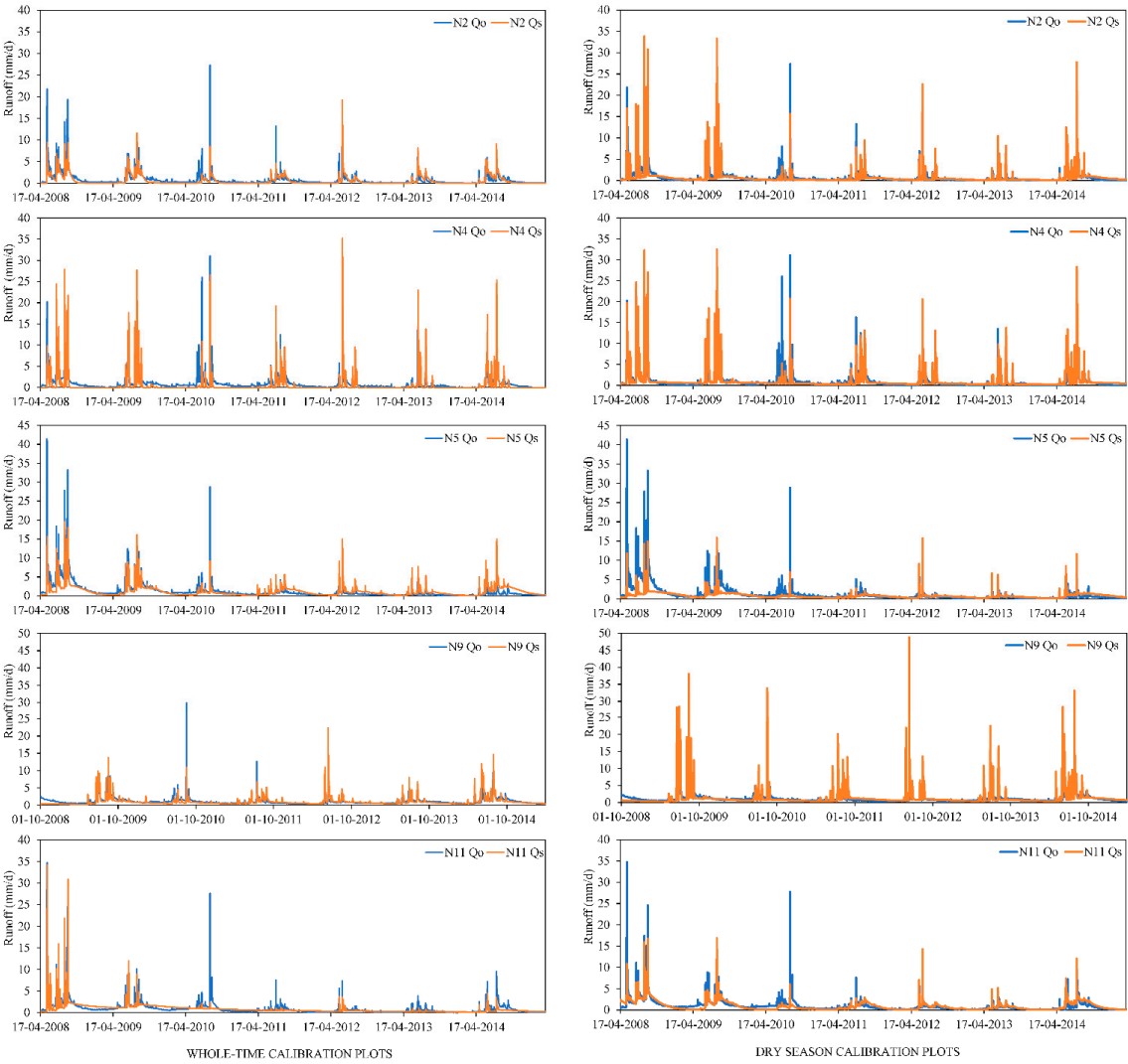

**Figure 3.** Simulated ($Q_s$) and observed ($Q_o$) runoff for the different catchments (i.e., additional supporting information on all catchments can be found online in the Supporting Information section at the end of the paper, Figure S2).

### 3.2.2. Dry Season Series Calibration

Using the *dry season* calibration, the model presented a relatively lower predictive potential (not satisfactory), with a general overestimation of the discharges for the whole study period (Figure 3); however, the hydrograph shape was correctly reproduced for drier months, despite the simulated values being overall higher than the observed runoffs (MW test and KW test, $p = 0$). The NSE varied between $-0.2$ and $0.6$ for the calibration period, with an $NSE_{mean} = 0.4$, while it ranged between $-17.3$ and $0.3$ for the validation period, with an $NSE_{mean} = -3.1$ (Table 3).

### 3.3. Simulated Runoff under Land-Use Scenarios

### 3.3.1. Total Runoff Yield

The results of runoff yield (RY, mm) for the three different land-use scenarios and input data whole-time series calibration showed that: (1) overall, the simulated values were significantly lower than the observed ones; (2) under the current cover, simulated RY was significantly lower in six of the catchments (N2, N4, N5, N7, N9, N11) and higher in two (N3, N8); (3) under the partial harvest, simulated RY was lower in two catchments (N2, N7), higher in five catchments (N3, N4, N5, N8, N11) and similar to the observed values in N9; and finally, (4) under the native forest, simulated RY was lower than that observed in six catchments (N2, N4, N5, N7, N9, N11) and higher in the remaining two (N3, N8) (Table 4).

**Table 4.** Observed ($Q_o$) and simulated ($Q_s$) runoff yield under land use scenarios (% of change) for the two calibration procedures (i.e., Additional supporting information on Table 4 can be found online in the Supporting Information section at the end of the paper, Table S4).

| Catchments | *Whole-Time Series* **Calibration** | | | *Dry Season* **Calibration** | | |
|:---:|:---:|:---:|:---:|:---:|:---:|:---:|
| | ($Q_o$~$Q_s$) Current cover % | ($Q_o$~$Q_s$) Partial harvest % | ($Q_o$~$Q_s$) Native forest % | ($Q_o$~$Q_s$) Current cover % | ($Q_o$~$Q_s$) Partial harvest % | ($Q_o$~$Q_s$) Native forest % |
| N2 | $-24$ | $-5$ | $-24$ | 0 | 68 | 2 |
| N3 | 8 | 10 | 9 | 48 | 79 | 49 |
| N4 | $-21$ | 8 | $-19$ | 30 | 96 | 32 |
| N5 | 0 | 29 | $-29$ | $-32$ | $-10$ | $-32$ |
| N7 | $-30$ | $-30$ | $-30$ | 96 | 166 | 96 |
| N8 | 19 | 12 | 19 | 106 | 152 | 106 |
| N9 | $-1$ | 4 | $-2$ | 33 | 51 | 34 |
| N11 | $-9$ | 50 | $-18$ | $-8$ | 29 | $-19$ |

The runoff yield was subsequently simulated using the *dry season* calibration, and the results indicated that (1) overall, simulated values were significantly higher than the observed ones; (2) under the current cover, simulated RY was significantly lower in three catchments (N2, N5, N11) and higher in five (N3, N4, N7, N8, N9); (3) under the partial harvest, simulated RY was lower in one catchment (N5, N7), but higher values in the remaining seven; and finally, (4) under the native forest, simulated RY was again lower in only one catchment (N11) but was higher in the other seven (Table 4).

### 3.3.2. Daily Runoff

Despite the observed variability between catchments, the simulated daily runoff ($Q_s$) was overall close to that of the observed ones ($Q_o$) (Additional supporting Table S5). The simulated higher runoffs (i.e., $Q_5$) were generally overestimated, especially in N2, N9 and N11, whereas N5 was greatly underestimated compared to the observed runoff (regardless of calibration procedure and land use):

(1) Using the *dry season* calibration, simulated $Q_5$ runoffs were higher (mean = 4.48 mm/day) than those obtained using the *whole-time series* calibration (mean = 3.33 mm/day) and, in any case, both were higher than the observed runoff (mean = 2.84 mm/day); (2) Regarding land use, simulated $Q_5$ was higher than the observed values, especially for partial harvest (3.81 mm/day).

The simulated median runoffs (i.e., $Q_{50mean-s}$ = 0.50 mm/day) were almost identical to the observed ones ($Q_{50mean-o}$ = 0.52 mm/day), despite some differences that are understandable when calibration and land uses are taken into account: (1) in general, the simulated values using *whole-time series* calibration were slightly higher than the mean, especially in the case of N8, whereas runoffs were underestimated in N2, N3 and N4 (the model yielded no flow) and N7. Results following the *dry-period* calibration showed a higher degree of fit between simulated and observed values, with the model yielding no zero runoff in any catchments; (2) as in the case of $Q_5$, simulated median values under partial harvest scenario yielded much higher values (5.16 mm/day) that those observed in the field. The simulated low runoffs ($Q_{95-s}$) were much higher than the observed ones (0.15 mm/day), regardless of the calibration procedure used (i.e., 0.14 mm/d in the case of the *whole-time series* calibration and 0.28 mm/d using the *dry-period* calibration procedure). Notably, both N11 and the 0-runoff simulations for catchments N2, N4 and N7 were overestimated in the *whole-time* calibration. Once more, the simulated runoffs under partial harvest were much higher than the observed ones, especially in the case of the *whole-time* calibration (0.18 mm/day).

Finally, no significant differences occurred when comparing Q5, Q16, Q50, Q84 and Q95 daily runoff between flow observed and simulated in each group (MW or KW tests), indicating similar hydrological behaviour with regards to the minimum, maximum or median runoff of each catchment.

Using the whole-time series calibration, the simulated median runoffs in the smaller catchments (N2, 13.9 ha; N3, 7.1 ha; N4, 7.6 ha; N5, 14.2 ha; and N7, 16.9 ha) were lower than observed runoffs, and were always lower under land current cover and native forest than under partial harvest (which represents the lowest forest cover of all land uses). In the larger catchments (N8, 54.9 ha; N9, 98.3ha; and N11, 413.6 ha) the simulated runoffs were always lower under land current cover than in partial harvest and native forest.

The runoff standard deviation (RSD) indicates the variability of stream runoff around the median which, in the case of the studied catchments, was much lower in the case of the observed discharges (7.5) than for the simulated ones (11.5). Overall, the *whole-time series calibration* showed higher variability (13.7) than the *dry-period calibration* (9.3), especially in light of the extraordinary runoff variability in catchment N3. Excluding N3, the simulated values obtained for the rest of the catchments show even less runoff variability for current cover and native forest than the observed runoff (5.8 and 6.1, respectively).

*3.4. Runoff Simulation under Climate Scenarios*

Figure 4 shows simulated runoff under different climate scenarios. The modelled values show a similar tendency to that of the simulated runoff under the *scenario-neutral approach*. For climatic scenarios in which rainfall varied between −20 and 0%, and evapotranspiration varied between −20 and 20%, the simulated runoff was 14 to 20% lower than those of the *scenario-neutral approach* in small catchments: N2 (13.9 ha), N3 (7.1 ha), N4 (7.6 ha) y N7 (16.9 ha). In contrast, when the rainfall ranged between 0 and 20% and the evapotranspiration fluctuated between −20 and 20%, the simulated runoff was 10 to 32% above the simulated runoff for the *scenario-neutral approach* for N5 (14.2 ha), N8 (54.9 ha), N9 (98.3 ha) and N11 (413.6 ha).

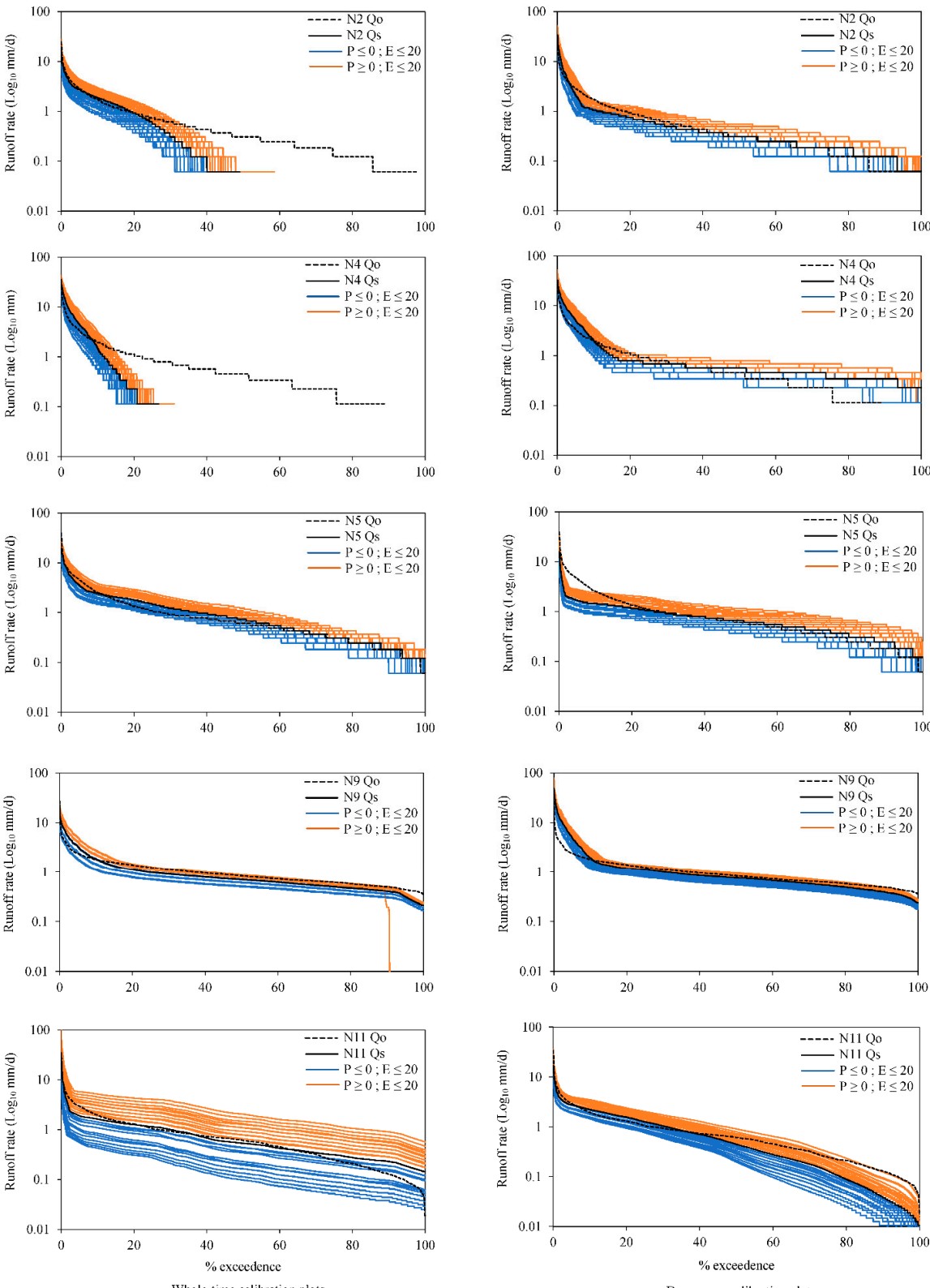

**Figure 4.** Flow duration curves on the four catchments' groups, flow duration curves were developed using daily data (i.e., additional supporting information on all catchments can be found online in the Supporting Information section at the end of the paper, Figure S3).

For climatic scenarios where evapotranspiration increased 20%, the runoff yield reduced than those of the scenario-neutral approach (Figure 5) in catchments N2 (13.9 ha), N4 (7.6 ha), N5 (14.2 ha) and N11 (413.6 ha). For climatic scenarios in which evapotranspiration decreased 20%, the simulated runoff yield showed a remarkable increase in catchments N3 (7.1 ha), N7 (16.9 ha), N8 (54.9 ha) and N9 (98.3 ha).

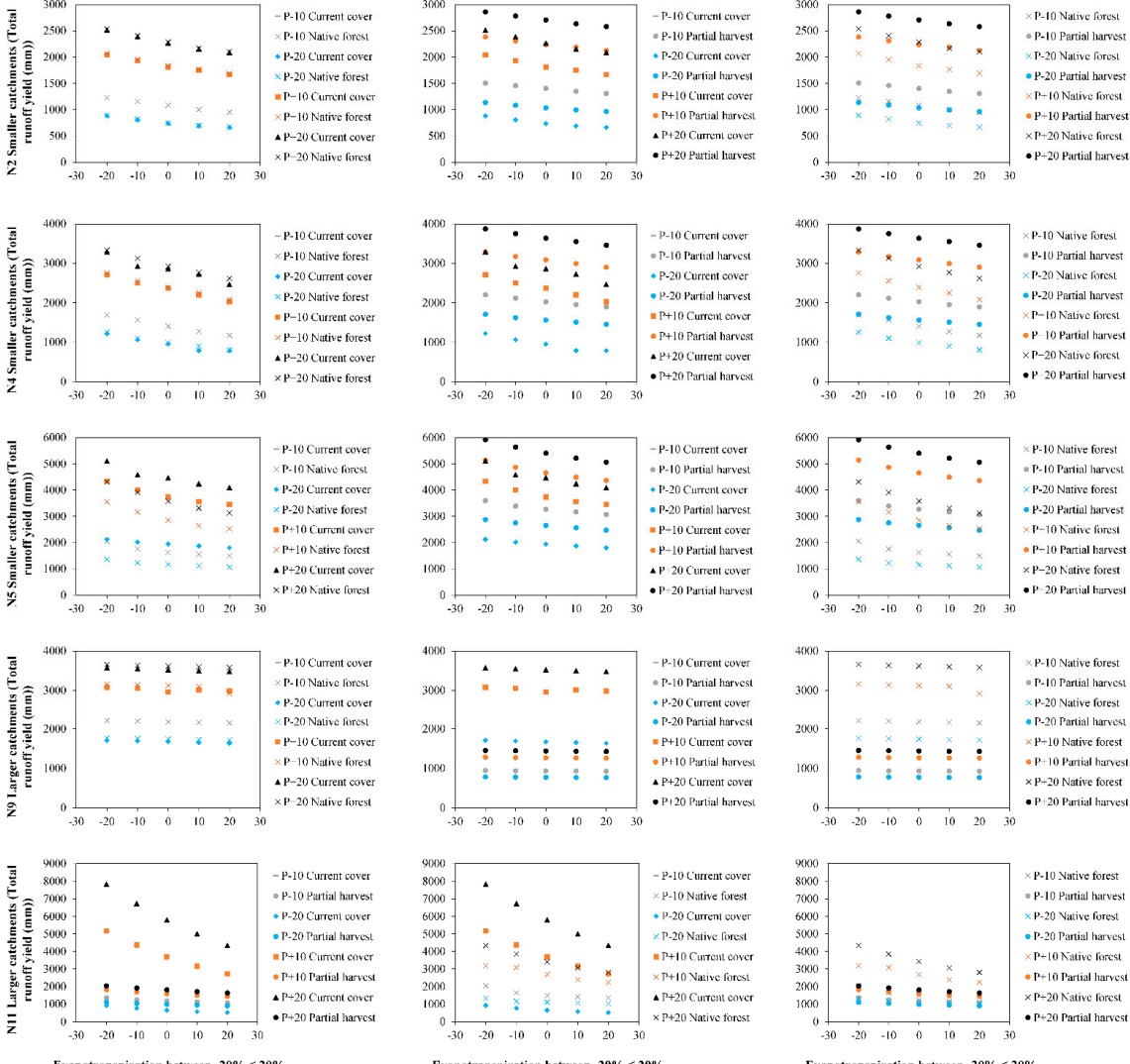

**Figure 5.** Runoff yield associated with different climate scenarios and land uses for the different catchments under study (i.e., Additional supporting information on all catchments can be found online in the Supporting Information section at the end of the paper, Figure S4).

### 3.5. Simulated Runoff under Land Use and Climate Scenarios

The highest runoff yields were obtained for catchments of group 4, specifically in (i) catchment N8 under land-use scenarios current cover and native forest, and (ii) catchments N9 and N11 for scenario partial harvest. Catchment N2 showed similar runoff yield under current cover and native forest, whereas N5 showed a decrease in runoff yield when current cover and native forest were applied and an increase under partial harvest. The driest climate scenario (20% decrease in rainfall and 20% increase in evapotranspiration), predicted a high loss of runoff for the modelled period (Figure 5); while the wettest scenario (20% increase in rainfall and 20% decrease in potential evapotranspiration) predicted 50% higher runoff yield than the one obtained for the high runoff losses (*driest*) scenario.

However, the catchments showed a similar trend ($p < 0.05$) in runoff yield for use scenarios current cover and native forest (Figure 5). In contrast, we found different significant runoff yield trends in the scenarios of current cover and partial harvest (Figure 5).

## 4. Discussion

### 4.1. Effects of Land Use and Climate Scenarios

The observed and simulated runoff yields are similar with results recently obtained in other Mediterranean catchments (in this case in the northern hemisphere, e.g., [6,41,42,45]. These authors included land uses and climatic scenarios similar to those used in our study.

The time series analysed in this study provided no statistical evidence that rainfall decreased during the study period in the Nacimiento area (1 April 2008 to 31 March 2015; [24]). It is noteworthy that overall, annual rainfall and annual runoff have been reduced in the region due to the mega-drought that Chile has experienced since 2010 [12]. The simulated runoff yield showed better agreement with the observed values when using the dry season calibration, overall reducing the number of catchments that demonstrated a very low runoff yield with whole-time calibration. Similarly, greater changes in simulated daily runoff were found in all catchments when the *whole-time series* calibration was used, the higher runoff being the most affected (general reduction). On the other hand, under the 20%-less precipitation scenario, daily runoff was not seemingly affected (both in magnitude and duration) decreasing between 14 and 20%, however, climate change scenario model Coupled Model Intercomparison Project (CMIP) [12] indicated that this region flow would decrease by approximately 40% by the end of the century.

A relationship between changes in flow duration and land-use scenarios was not found (i.e., hydrological response between catchments is highly variable), initially suggesting that, in general, observed future changes in river runoff may be mostly related to climate variability rather than changes in land use/cover.

The driest climate scenario (20% decrease in rainfall and 20% increase in evapotranspiration) predicted a 20 to 61% decrease in runoff yield, while the wettest (20% increase in rainfall and 20% decrease in potential evapotranspiration) predicted an increase in runoff yield by 21 to 55% over the scenario-neutral. However, these effects were not seen in all catchments, suggesting that the reduction or increase in runoff yield was also driven by other factors. As stated, flow patterns are also altered by changes in land use (e.g., [22,46,47]. In most of the Nacimiento catchments, runoff yield increased with the reduction of forested area (i.e., partial harvest), whereas changes associated with current cover and native forest were much less evident and variable between catchments. The runoff was also altered by size of the catchment, reducing to more than half under a 20% decrease in rainfall and a 20% increase in evapotranspiration in small catchments.

Our modelling results suggest that land use and climate affect catchment runoff, but the precise role of each of them is still difficult to specify, especially in the smaller catchments. For instance, the classic work by Bosch and Hewlett [19] and the more recent study by Best et al. [20], Andréassian [21] and Brown et al. [22], predicted that in general, the establishment of forest plantations on old agricultural lands, and the forest growth, affect runoff by reducing the flow rates and the runoff yield, especially during the dry season (see more examples in [15,18,23]. Several field studies showing greater forest cover leads to higher flows over the lower part of the runoff duration curves [48–51]. This research does not show that greater forest cover corresponded to higher runoff over the lower part of the RDC. However, it suggests that lower forest cover (partial harvest) leads to higher runoff under the "dry season calibration". This increase is the result of a better representation of peak runoff, and the partial harvest of forest led to greater runoff via reduced evapotranspiration [51–54].

Kreutzweiser et al. [55] and Jinggut et al. [56] emphasized that the understanding of hydrological processes is particularly important in ecosystems where intensive land use can further increase the effects of climate change. Climate change will likely lead to an *intensification* of the hydrologic cycle

in areas where water use by vegetation is currently limited, but elsewhere, can lead to a net drying effect [57–61] predicted that regardless of the cover of a treated catchment, extreme climatic variability remains the more dominant driver of flow response.Implications drawn from these analyses, which can be relevant to water resource management in forested catchments are (a) the need for continuous monitoring over long periods [61,62], especially in smaller catchments which are more sensitive to significant driest climatic scenarios; (b) smaller catchments affected by driest climatic scenarios yield higher runoff under land partial harvest (lower forest cover); (c) larger catchments under all climatic scenarios yield higher runoff under land partial harvest and native forest; and finally, (d) the knowledge of potential climate change effects may eventually facilitate sound forest management decisions and minimize the impact of forest treatment, avoiding particularly problematic weather conditions [63,64]. The contrast between the climatic scenarios is so great that, whatever the forest cover and catchment surface, the climatic variability [14] will continue to be the most dominant driver of the flow responses at sites characterized by a Mediterranean climate with dry summers.

## 4.2. Model Limitations

The limitations of the model arise mainly from the structure of the mass balance equations, which describe the hydrological differences at different spatial scales. This relates, for example, to land cover (e.g., percentage of vegetation) and soil characteristics (i.e., permeability). The performance of these variables is affected by the type of input information and the scale of available raster cells. Based on the size of the studied catchments and the nature of the equations involved in the simulations, it may well be that the spatial resolution of input data is most definitely not enough to represent the variability in catchments. Therefore, the spatial resolution at which the model was run was a compromise between the available information, the iteration time, the number of sites and the physical nature of the processes reproduced by the model.

Regarding the description of the soil, the model considers the vertical and horizontal permeability of the soil surface and the aquifer as separate input information, considering both the permeability of the parent material under the aquifer and the potential runoff losses [6]. Complete, detailed information to reproduce the spatial variability of permeability processes was not available for this study, so data input was prepared utilizing an interpolation process based on statistical models that include autocorrelation between midpoints. These methodological shortcomings may affect the way the model reproduces the dynamics of water in the soil. The model does, however, consider the vertical permeability of clay soils that directly affect water infiltration and runoff processes (i.e., the rest of the permeability parameters are less relevant for this study). Herrero et al. [6] argued that discharge underestimation might reflect an overestimation of soil permeability. Furthermore, vertical permeability at the soil surface is the parameter controlling infiltration in the water mass balance equation used by the model, and, for example, a soil classified as sandy may have a permeability that varies over three orders of magnitude. Moreover, the model was calibrated by maximizing the NSE coefficient, though it must be noted that high flow episodes correspond to a small fraction of the data period (Figure 3). Therefore, a calibration for the dry period (i.e., *dry season* calibration) improved the relationship between observed and simulated data for mean $Q_s$, while higher differences were still observed for peak discharges (Figure 3). In addition, the underestimation of the peak $Q_s$ can lead, in turn, to a decrease in the simulated total runoff.

Interpolation was also used to prepare meteorological data files used as model inputs. The use of the Hargreaves and Samani [28] equation to estimate evapotranspiration may have also added bias to model calculations, as an overestimation of evapotranspiration likely decreases water storage in the soil, which in turn decreases the availability of dynamic storage in the catchments [24]. The influence of temperature is also an important factor in areas with a Mediterranean, subtropical climate. Dry summers, together with extreme temperatures, affect water consumption through evapotranspiration, decreasing water reserves in the soil, altogether affecting how the model reproduces surface runoff.

Finally, another factor that may influence model performance is including current cover (in which the canopy cover is higher than in the other two land-use scenarios) as data input for TETIS®, doing so may cause $Q_s$ runoff to reflect higher interception and evapotranspiration (e.g., [24,65], a fact that could explain the lower $Q_s$ peaks compared to the observed ones, especially during the dry season. However, as stated, it is important that not all $Q_s$ peaks were underestimated by the model (Figure 3), so differences may develop from more complex interactions between factors such as altitude, soils, geology, rainfall and temperature distribution that are particularly important in Mediterranean climates, and whose particular roles and contributions are likely not yet fully understood.

All these restrictions affect the results in specific ways. For (1) the spatial resolution of information, the model is affected by not reproducing the water dynamics of the subsoil in its entirety; (2) for the NSE coefficient and maximum underestimated runoff, the model is affected by quantifying a decrease in the simulated total runoff. Krause et al. [66] found that the NSE is sensitive to extreme values due to squared differences (to overcome cases of extreme values, they recommended the use of logarithmic and relative derivative forms of NSE); (3) the potential evapotranspiration equation of Hargreaves and Samani [28] affects the model by quantifying an overestimation of evapotranspiration, decreasing the storage of water in the subsoil [24];(4) with regard to temperature, the model is affected by quantifying extreme temperatures, completely altering the way the model reproduces surface runoff and water consumption through evapotranspiration; and (5) regarding land use, the model is modified by quantifying less or greater interception and evapotranspiration, due to differences in coverage size over the years, resulting in modelling lower than observed runoff.

## 5. Final Remarks

The application of the TETIS® hydrological model was considered satisfactory since the model was able to reproduce the annual distribution, at a daily resolution, of the streamflow that is mainly associated with rainfall events between April and September (the winter season in Chile). We found that land use and climatic changes have contrasting effects on runoff (runoff yield): (i) smaller catchments affected by the driest climatic scenarios experienced higher runoff when the forest cover was lower than under full forest cover (plantations or native forests); (ii) in contrast, larger catchments under all climatic scenarios yielded higher runoff below the full forest cover than under partial harvest and native forest. In particular, the highest runoff yields were obtained for catchments N8, N9 and N11 under land-use scenarios of partial harvest and native forest. Catchment N2 showed a similar runoff yield under current cover and native forest, similarly to N3, N4 and N5, whereas N7 showed a decrease in runoff yield under native forest and an increase under partial harvest. In turn, climatic scenarios uniformly affected the hydrological response of the catchments, independently of their size; the driest climate scenario predicted a high loss of runoff for the modelled period, while the wettest situation predicted higher runoff yield, i.e., 50% higher than at scenario-neutral. Overall, the results suggest that, in general, future changes in runoff can be mainly related to a higher influence of climatic variability rather than changes in land use.

This study proved to be relevant to support already ongoing discussions related to forest management, which are intended to minimize the impact of forest cover and climate change on a basin's runoff yield. The study focuses on small catchments, as they are likely to be especially affected by land-use and climate changes.

The use of TETIS® showed the potential of numerical models. The analysis of hydrology at the catchment scale provided useful tools for catchment management and associated runoff resources. However, the modelling exercise implies some limitations that are summarised here:

(i)　Land use: the land use maps of 2015 were used as input data of the model for the reproduction of the simulations. It is likely that the forest cover in 2015 was greater than the average coverage between 2008 and 2014; therefore, the simulated results correspond to conditions enhancing interception and evapotranspiration.

(ii)　NSE coefficient: the underestimated maximum runoff corresponded to a small fraction of the data period, but even these can lead to underestimations of the total volume of runoff.

(iii)　Meteorological data: the resolution and extension of meteorological data used as input to the model should be improved. A network of weather stations would be necessary to correct errors; for example, the influence of altitude on rain and temperature can be particularly important factors in highly contrasted areas such as those in the Mediterranean region.

**Supplementary Materials:** The following are available online at http://www.mdpi.com/2073-4441/12/6/1828/s1, Figure S1:Additional supporting information of the three land cover scenarios, Figure S2: Additional supporting information of the Figure 3 (all catchments), Figure S3: Additional supporting information of the Figure 4 (all catchments), Figure S4: Additional supporting information of the Figure 5 (all catchments), Table S1: Summary of hydroclimatic scenarios with selected increments of precipitation and potential evapotranspiration used on modelling exercises (± for reference standard deviation is also shown), Table S2: Modelling results for each of the studied catchments together with forest management and physiographic information, Table S3: Characterization of catchments hydrology, Table S4: Observed ($Q_o$) and simulated ($Q_s$) runoff yield under land use scenarios (% of change) for the two calibration procedures.

**Author Contributions:** G.B. and A.H. contributed to the project idea development, methodology and modelling validation, prepared the manuscript (writing the original draft, reviewing and editing, collecting and preparing the figures); A.I. supervised the research, revised-editing the manuscript and project administration; R.J.B. supervised the research and revised-editing the manuscript; O.M. project administration. All authors have read and agreed to the published version of the manuscript.

**Funding:** This research received no external funding.

**Acknowledgments:** The Graduate School of the Faculty of Forest Sciences and Natural Resources, Universidad Austral de Chile, is thanked for its support and infrastructure. The authors acknowledge the support PhD scholarship from Comisión Nacional de Investigación Científica y Tecnológica (CONICYT). The authors acknowledge the support from the Economy and Knowledge Department of the Catalan Government through the Consolidated Research Group 'Fluvial Dynamics Research Group'—RIUS (2017 SGR 459).

**Conflicts of Interest:** The authors declare no conflict of interest.

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
