# Peer review of "Modelling the Effects of Changes in Forest Cover and Climate on Hydrology of Headwater Catchments in South-Central Chile"

_water, doi:10.3390/w12061828_

Round 1

Reviewer 1 Report

General comments

While the topic of this study is of relevance to this journal and of interest to the broader community, the manuscript itself is very long, and the study lacks sufficient information on model parameterisation and calibration. Analysis and interpretation is difficult to follow, and provides description of the results without really providing any meaningful insights into the relationships between climate change, forest cover and streamflow in Chile. The study reaffirms that there are many factors (that are not included in this modelling study) that make it difficult to draw quantitative conclusions, and that climate change will have a larger impact on catchment hydrology than changes in vegetation cover.

I have included a PDF version with annotated edits, comments and suggestions. I feel this manuscript is not of a high enough standard and should only be considered for publication in Water with major revision. The manuscript would be improved with a review by a native English speaker.

Some of the main comments and suggestions are:

  1. I suggest avoiding attaching qualitative or emotional words to scenarios. eg. increases in temperatures may be good for some people/land uses. It is unclear what a pessimistic scenario is. I prefer to use direct language eg. scenarios in which rainfall was lower and ET was higher, or the wettest and warmest scenario etc.
  2. There are comments made about the changes in vegetation eg "higher transpiration is caused by species composition…" without any clarification of what is meant by this. The species of eucalypt within the plantations (the main land use focus of this study) is not mentioned. It is not clear if the pine and eucalypt plantations were parameterised differently. There were large differences in plantation stand density between some of the catchments. How was this taken into account?
  3. Canopy interception is the only vegetation parameter mentioned in the study – but no data is provided. How was vegetation represented in the model? How did you parameterise each of the land use types in the model?
  4. Measured information on plantation/vegetation (ie biomass, density, age...) is provided but does not appear to be used in anyway during the model runs nor is it referred to in the text.
  5. The data set used from 2009-2015 is very short (possibly too short) to be effective in calibration and validation of a model which is then used to investigate changes/trends that occur over much longer timescales.
  6. It is unclear at what spatial resolution the input data was, and what scale the model water run, and at what time scale the model outputs were generated. This is all basic information that should be included in any modelling study. There is also no detail on what time step NSE was calculated. eg. model was run on a daily time step and NSE calculated on monthly totals?
  7. Given that the model is most sensitive to rainfall inputs, It is important to provide detail on which method was used to interpolate/extrapolate rainfall and temperature measurements across the study region and catchments. It seems that rainfall and temperature were input into the model at a 10km2 resolution, which if true, is much too coarse for this study, as the study catchments are much smaller than this.
  8. Only NSE is used to optimise and assess the models. I suggest also including values of mean and standard deviation of the model runs compared with the observed data to see whether the models have the total volumes and variation about right.
  9. The tables are very cluttered and quote annual values in mm to the first decimal point. I suggest quoting to the nearest mm and removing standard deviations.
  10. I am unsure why the authors have decided to express streamflow output in daily flow per second, rather than per day? Also, it is unclear why flow is expressed as volume rather than as a depth (to normalise for catchment area)? Many of the differences in the results arise from differences in catchment size when comparing flow volumes that do not normalise for catchment size. Water yield is expressed as a depth or change in %. Otherwise it is difficult to compare between catchments. The plots themselves make it impossible to compare scenarios.
  11. Some statements are vague and offer no insight. For example ….. when ET increases, flow will decrease. This does not need to be stated unless you can provide quantification. Statements such as … when there is more rain, there is more flow ..... or when ET is reduced, flows increases, without any quantification is not insightful.
  12. Using scenario numbers in the text makes it difficult to read. I suggest relating the change in tree cover % of these scenarios or using total tree cover in each scenario to label them. Also, instead of group 4, please talk about catchments between 55-420 ha etc......
  13. The discussion claims this study helps managers with the treatment of plantations, but this study did not simulate or discuss the treatment or management of forests at all. It simply looked at plantations as a % of catchment area.
  14. It appears that flows during the dry season are more important to estimate, yet there is very little information provided as to why this is so, and what the impacts of low dry season flows have on the community and environment.
  15. It is very difficult for the reader to understand how the changes in climate and land cover interacted and what the effect in streamflow was from these simulations. I suggest the authors remove the large tables with many numbers and replace them with some well thought figures that summarise what is the main focus of this study.

Author Response

Reviewer 1

LINE 26: I suggest avoiding attaching qualitative or emotional words to scenarios. eg. inncreases in temperatures may be good for some people/land uses. It is unclear what a pessimistic scenario is. I prefer to use direct language eg. scenarious in which rainfall was lower and ET was higher .......

Answer:

Reviewed and changed

LINE 30: This is a vague statement and it is unclear what you mean.

Answer:

Reviewed and changed

LINE 43: reference?

Answer:

Reviewed and changed

LINE 47: unsure what this means. Do you mean water yield into storages?

Answer:

Reviewed and changed

LINE 49: Please describe what you mean by dry. Also, which season is dry?

Answer:

Reviewed and changed

LINE 49: Which species?

Answer:

Reviewed and changed

LINE 50: What do you mean by this type of water shortages? Are you referring to water storage in reservoirs?

Answer:

Reviewed and changed

LINE 63: I think you mean an increase in the proportion of catchment area that is forested

Answer:

Reviewed and changed

LINE 64: Higher transpiration is caused by species composition? This does not make sense. Also, are you saying that transpiration continues to increase with age?

Answer:

Reviewed and changed

LINE 66: What do you mean by forest use?

Answer:

Reviewed and changed

LINE 76-77: it's unusual to use the entire dataset for calibration, as there is no independent data then left for validation

Answer:

Reviewed and changed

FIGURE 1: Please provide a legend item for the grayscale on the maps used for elevation.

Answer:

Reviewed and changed

LINE 105: Please provide a general description of this soil type eg. textures and depths

Answer:

Reviewed and changed

LINE 112: Please include details of the species of eucalypts

Answer:

Reviewed and changed

TABLE 1: Was this information on vegetation (ie biomass, density, age...) used in anyway during the model runs? If not, why is it mentioned here?

Answer:

They were not used in the model inputs but were used to group the catchments Table 4.

TABLE 1: Do you mean average annual daily flows?

Answer:

Reviewed and changed

LINE 122: At what resolution? eg. every minute?

Answer:

Time series of temperature, rainfall and runoff were directly obtained in the field, were recorded with a 6-minute resolution and summarized as daily values

LINE 123: Is this the resolution at which is was measured? or at which analysis was done?

Answer:

A daily resolution for analysis was adopted

Table 2: Please define the hydrological year

Answer:

Considering Hydrologic Water Year (April 1st to March 31st)

TABLA 2: For rainfall and ET, I don't think it is reasonable to quote to the first decimal place. Please quote to whole mm. How did you derive standard deviation for the annual rainfall when you are providing one number for one year?

Answer:

Standard deviation σ (sigma) is the square root of the variance of X; i.e., it is the square root of the average value of (X − μ)2.

LINE 138: do you mean soil water storage? ie. including all pores?

Answer:

Soil water storage, including; static storage, Surface storage, Gravitational storage, Underground storage, Initial state of the aquifer, Initial state of the interception tank.

LINE  139: what do you mean by water losses? ET?

Answer:

Water losses (by evapotranspiration)

LINE 140: although you did not include snowmelt in this study. a little confusing to mention it here.

Answer:

Reviewed and eliminated

LINE 141: Canopy interception is the only vegetation parameter mentioned here. How was vegetation represented in the model? how did you parameterise each of the land use types win the model?

Answer:

There are two files related to the effect of evapotranspiration: the land cover codes (cobveg) and the Vegetation Factor or temporal distribution of the ET factor (FactorETmes.txt).

The coverage index for Evapotranspiration modeling is calculated from the type of vegetation cover, based on a reclassification of the vegetation / land use map.

Each cell is assigned a numerical value corresponding to the vegetation type. An annual variation of evaporation is assigned to this numerical value in the vegetation factor table (ETmes).

The vegetation factor (FAO crop factor) represents the variability of evaporation in the annual cycle. It indicates the growth status or the amount of biomass of each type of vegetation. For each category defined in relation to the vegetation cover, an evapotranspiration factor value is assigned. The values ​​have been obtained from FAO Report 56: "Crop evaporation: Guides for determining crop water requirements" (fao.org).

The ETmes factor file has 14 columns: 12 correspond to the evapotranspiration values ​​of each month, the penultimate is the interception value of the vegetation, which is also included in this file (later the interception is explained), and the last corresponds to the vegetation codes used. Columns always separated by spaces. This file does not have headers of any kind, for example:

* Example

1.00 1.00 1.00 1.00 1.00 1.00 1.00 1.00 1.00 1.00 1.00 1.00 9 01- Evergreen Forest

0.60 0.60 0.80 1.00 1.00 1.00 1.00 1.00 1.00 1.00 0.80 0.60 9 02- Deciduous forest

0.60 0.60 0.60 0.60 0.60 0.60 0.60 0.60 0.60 0.60 0.60 0.60 1 03- Perennial arboreal crop

LINE 148: purposes such as?

Answer:

Reviewed and added: impact of climate change on, runoff trend and sediment yield

LINE 153: Given that the model is most sentitive to rainfall inputs, would will have to provide more detail on which method you used to interpolate/extrapolate rainfall and temperature measurements across the study region.

Answer:

Reviewed and added: Kriging interpolation was used, calculating the square difference between the values of the associated locations (Heine, 1986).

LINE 153: Does this mean that rainfall and temperature were input into the model at a 10km2 resolution? Your study catchments are much smaller than this

Answer:

Reviewed and changed

LINE 175: Please provide more detail on the ground truthing eg. how many sites were used for validation? was it easy to distinguish between native forests and plantations?

Answer:

The identification of planted tree species was performed by image recognition, supported by ground-truth validations in the field, six points were used for validation on both sides of the streams per catchment, using monitoring transects 10 meters wide and 20 meters long, outside the riparian zone.

LINE 176: at what resolution?

Answer:

Reviewed and added 5-meter resolution

LINE 183: Parameters themselves have uncertainty. I don't think introducing them is to decrease uncertainty.

Answer:

The production model and the translation model presented in TETIS include some correction factors that globally correct the different parameters, allowing a quick and agile calibration (manual or automatic) of the different processes represented, taking advantage of the information used in their previous estimation.

These correction factors try to correct the following errors:

- Initial estimation of the parameters

- In the input hydrometeorological information to the model

- In the model itself

- Spatial-temporal scale effects

LINE 190: These are the calibration and validation results, so they really should be in the results section. Also these figures do not show NSE results as suggested in this sentence.

Answer:

The calibration and validation results are an important part of the methodology adopted in the analysis of this study, due to the low goodness of fit between simulated and observed discharges during the dry periods of the data series.

LINE 193: This is confusing. It seems that you used 2009-2012 data for calibration, and then 2012-2015 data for validation. is that correct?

Is correct, you used 2009-2012 data for calibration, and then 2012-2015 data for validation.

Answer:

First calibration: discharge series 1 April 2009 to 31 March 2012 was considered for the computation of NSE.

Second calibration: excluded wet season data (April to September), series between 1 April 2009 to 31 March 2012 for the calculation of NSE.

LINE 195: It would be good to provide more detail of this importance in the introduction (than you already have included).

Answer:

Added to the introduction on the assessment of water resources during the dry period: This approach was adopted to obtain reproduction of discharges during the dry season, as the assessment of water resources during this period is the main focus of this study due to its importance from ecological and socio-economic standpoints

LINE 200: Please include information on at what time step the model was run and at what time step NSE was calculated. eg. model was run on a daily time step and NSE calculated monthly

Answer:

The NSE was applied to test the quality of the simulated results, on a daily time step and end of each data series, using the following rating criteria

LINE 206: Looks like daily data. Please confirm.

Answer:

On a daily time step

LINE 208: Please reconsider how you call this. It is confusing to say 'whole series calibration' when data from 2009-2012 was used for calibration.

Answer:

It was reviewed and it was decided to keep the names assigned to the two calibrations

LINE 215: Bare as in no vegetation cover at all? Which land use category is this. 'harvested'? how realistic is this?

Answer:

Partial harvest: consider the use of land after the partial harvest of 50% of the catchment area

Realistic condition, forest management in Chile considers clear cutting of forests planted with a maximum of 300 ha continuous in at least 70% of the surface harvested annually, allowing exceptions not exceeding the legal maximum, in areas not near main routes, areas of relevant tourist interest or exposure to centers with more than 50,000 inhabitants.

LINE 217: It seems that representation of land use in the model is static ie. does not change during a model run. Please confirm.

Answer:

The land use in the model is static.

LINE 236: Showing SD here is not very helpful and only clutters an already busy table.

Answer:

Reviewed and eliminated

Table 3: Please quote annual totals to the nearest whole mm

Answer:

Reviewed and changed

LINE 242: These are the calibration and validation results and should really be in the results section along with figure 2.

Answer:

Reviewed, changed and created table 3 indicated NSE value of calibration and validation.

TABLA 5 There is a big difference in plantation density between group 1, 2 and the others. Is plantation density explicitly modelled? If so, how would this effect the results?

Answer:

The plantation density is not used in the model inputs, considered to group the catchments Table 5.

LINE 264: I think results should start off with the calibration and validation results

It was reviewed and it was decided to keep start of results.

Answer:

The calibration and validation results are an important part of the methodology adopted in the analysis of this study, due to the low goodness of fit between simulated and observed discharges during dry periods.

LINE 267: Need more detail of the sampling interval. Not sure how important this is. You do not model or discuss rainfall intensity

Answer:

Reviewed and changed in table 2

LINE 276. of all catchments? not unusual given the small size of every catchment.

Answer:

In all catchments the response was quick in relation to the size of each catchment

LINE 289: Which table are you referring to... table 4? That table does not show good agreement in terms of NSE. 4 of the 8 catchments had NSE<0.5 which is considered unsatisfactory in the definition you have adopted.

Answer:

Reviewed and changed: Using the whole-time series calibration and land use for the year 2015, mean Qs of each of the four catchments groups (defined in Table 5) showed satisfactory agreement with Qo in N2, N5, N9 and N11 (Table 3).

LINE 292: Probably not difficult to reproduce the shape given the catchments are very small.

Answer:

Hydrograph shape was overall correctly reproduced in N2, N5, N9 and N11, catchment area between 14, 14, 98 and 414 ha

LINE 298: Please provide more detail on how they are different if you feel this is important to mention.

Answer:

Reviewed and eliminated

LINE 305 This is generally to be expected.

Answer:

Reviewed and eliminated

LINE 307: Not sure what you mean by spatial validation. These are time series plots. This shows a chart for each catchment, and not catchment groups. Data is shown for the whole calibration and validation period combined - which is odd as you treated them separately.

Answer:

Reviewed and changed

LINE 312: you mean simulated using the parameters and input data used for this calibration run?

Answer:

Reviewed and changed

LINE 324 - TABLA 7: It is difficult to interpret this very large table. I suggest trying to represent this information graphically.

Answer:

Reviewed

LINE 327: I do not understand what this means "magnitude of stimulated daily flows relative to the duration ....."?

Answer:

Despite the observed variability between catchments, of the simulated daily flows (Qs) as overall close to that of the observed ones (Qo) (Tables 8 and 9).

LINE 331: I do not understand why you would want to compare simulated flows from the 2 different calibration methods.

Answer:

Due to the low goodness of fit between simulated and observed discharges during the dry periods of the data series

LINE 333: Why have you decided to express daily flows per second rather than per day? Also why have you decided to express flows as a volume rather than as a depth (to normalise for catchment area)?

Answer:

Regarding the observed time series that are input to the model: series of flows registered in m3/ s (Q)

LINE 353. It is not surprising that there are differences based on catchment size when you have compared flow volumes that do not normalise for catchment size.

Answer:

Regarding the observed time series that are input to the model: series of flows registered in m3 / s (Q).

It was not normalized, and the decision was made to group the basins by their size. Table 5

TABLA 9: for which catchment is this?

Answer:

Reviewed and eliminated

LINE 380: Please be more quantitative rather than only saying flows were higher than another.

Answer:

Reviewed and changed

LINE 382: I think it is obvious that when ET increases, flow will decrease. I don't think you need to say this.

Answer:

Enter the result associated with the ET

LINE 382: Reduced from what? from the calibration simulation?

Answer:

Reviewed and changed: For climatic scenarios where evapotranspiration increased 20%, the water yield reduced than those of the scenario-neutral approach

LINE 384: you are saying that the modelled flows in the catchments are similar to observed flows in the other catchments? I do not understand.

Answer:

Reviewed and eliminated

LINE 386: You do not need to include statements that when there is more rain, there is more flow..... or when ET is reduced, flows increases.

Answer:

Reviewed and eliminated

FIGURE 7: I do not understand why water yield is expressed as a volume rather than a depth or %. It means it is difficult to compare the catchments. The plots themselves make it impossible to compare scenarios.

Answer:

Reviewed and changed

LINE 404: How are they similar?

Answer:

Reviewed and changed

LINE 406: Which begs the question, why did you do this study if it has been done before?

Answer:

This study was carried out to evaluate the potential of the TETIS hydrological model for basins in the southern hemisphere that have climatic conditions like the sites of the studies by Prudhomme et al., 2010; Buendia et al., 2016; Bussi et al., 2016b; Herrero et al., 2017. But with soil use conditions other than intensive forest management

LINE 408: I don't think this is relevant given the short length of data available.

Answer:

The time series analyzed in this study provide statistical evidence for Chile, currently there are 10 years of data with a resolution of 6 minutes, this database is recognized and used by forest hydrology, clearly it does not compare with decades of databases from the northern hemisphere, but this study involves a large number of catchments under study.

LINE 412: Decreasing trend over what period? You have only provided data for 2008-2015 and just said there is not trend.

Answer:

Reviewed and eliminated

LINE 413: How was it related?

Answer:

Reviewed and eliminated

LINE 412: I do not understand what is being stated here.

Answer:

Reviewed and changed

LINE 416: I do not understand what duration means in this context.

Answer:

Reviewed and changed

LINE 418: I don't understand, daily flow did not changes when rainfall was decreased by 20%? really?

Answer:

Reviewed and eliminated

LINE 420: What was the % rainfall decline than would leads to a 40% reduction in flow? How does this compare to your results? Also, in a real word scenario, you'd expect rainfall to decline and ET to increase... so why are you comparing your rainfall only decline to another study?

Answer:

The percentage of decrease in rain is 20% and according to the study by Garreaud et al., 2017, the flow rates would decrease by 40%. In our study the flows decreased between 14 and 20%

LINE 426: This is fairly obvious. Please provide some quantitative analysis of this.

Answer:

Reviewed and changed

LINE 433: Reducing under what conditions?

Answer:

Reviewed and changed

LINE 435: I think this a well established process that does not require more modelling to confirm.

Answer:

Reviewed

LINE 445: It would be helpful to provide a chart showing rainfall and ET for your study region/catchments for each month to show the relative contribution of each to catchment hydrology in the region. eg. when are you study catchments water or energy limited?

Answer:

Table 2 provides annual values of pp and et for the catchments under study.

LINE 466: I would guess that the spatial resolution of input data is most definitely not enough to represent the variability in catchments. Models only represent the real world.

Answer:

Based on the size of the studied catchments and the nature of the equations involved in the simulations, it may well be that the spatial resolution of input data is most definitely not enough to represent the variability in catchments.

LINE 498: Are you suggesing that ET was over estimated by land use #1? Why?

Answer:

Use 1 the canopy cover is higher than in the other two land-use scenarios, may cause Qs flows to reflect higher interception and evapotranspiration.

LINE 521: do you mean monthly or seasonal? Even the simplest model can reproduce annual flows

Answer:

The application of the TETIS® hydrological model was considered satisfactory since the model was able to reproduce at least the annual distribution, at daily resolution of streamflow that is mainly associated with rain events between April and September (winter season in Chile).

LINE 524: Using group and scenario numbers in the text like this makes it very difficult to read and understand. Instead of group 4, please talk about catchments between 55-420 ha) etc......

Answer:

Reviewed and changed

LINE 524: Similarly, please don't use scenerio numbers as it is difficult to read. Relate the change in tree cover % of these scenarios here.

Answer:

Similarly, please don't use scenario numbers as it is difficult to read. Relate the change in tree cover % of these scenarios here.

LINE 533: But you did not simulate or discuss the treatment or management of forests at all. You simply looked at plantations as a % of catchment area.

Answer:

Reviewed and changed

LINE 533: Why? Might there be a shortage of flows during the dry season? please explain.

Answer:

The flow of the small basins used in this study during the summer season decreases considerably to zero

LINE 546: Errors in climate inputs are likely to be greater than the differences you observed.

Answer:

Reviewed

Reviewer 2 Report

Dear authors

Abstract

Has to be rewritten with focus on your own results.

I am impressed…a lot of data.

I have some headaches…..after analyzing this manuscript….

BUT I have a few remarks:

  1. The manuscript is 17-31% plagiarism and auto-plagiarism from some papers already published: you know them already, but I will give you some examples:

2% match (publications)

Guillermo Barrientos, Andrés Iroumé. "The effects of topography and forest management on water storage in catchments in south-central Chile.", Hydrological Processes, 2018

3% match (Internet from 21-Jan-2020)

https://link.springer.com/article/10.1007%2Fs11368-017-1684-6

etc…. You know better than me!

  1. The manuscript is too long. If the editor decides to accept it…no problem for me
  2. The graphical illustration is very good (elaborated)
  3. The methods are clearly presented
  4. The results are quite interesting. And I am very satisfied considering that are based on a large dataset. Impressive.....
  5. The Discussion section….is not very well structured… I would like to structure it according to results…. step by step analyzing all the aspects and relate them to other studies and experiments and highlight you own contribution.
  6. If you rewrite the paper more synthetized and avoid plagiarism, I recommend this manuscript for publishing. I am not blind, I can see a lot of valuable data here
  7. Check the bibliography…you have some inadvertences
  8. All the best

Author Response

REVIEWER 2

The manuscript is 17-31% plagiarism and auto-plagiarism from some papers already published: you know them already, but I will give you some examples: 2% match (publications)

Answer:

The similarity with the mentioned articles was reviewed. Indicate that the article Guillermo Barrientos, Andrés Iroumé "The effects of topography and forest management on water storage in catchments in south-central Chile.", Hydrological, is an article of which I am the main author, this article describes the study area used in this research and that is described in the methodology section.

The Discussion section….is not very well structured… I would like to structure it according to results…. step by step analyzing all the aspects and relate them to other studies and experiments and highlight you own contribution.

Answer:

Reviewed

If you rewrite the paper more synthetized and avoid plagiarism, I recommend this manuscript for publishing. I am not blind, I can see a lot of valuable data here

Answer:

Reviewed and changed

Check the bibliography…you have some inadvertences

Answer:

Reviewed and changed

Reviewer 3 Report

Overall, this work merits publication as the topic is of course of great interest and could be a good fit for the journal. The structure of the manuscript however should undergo some major reorganization as the enormous quantity of tables makes it very difficult to follow. Might I suggest submitting those as Supplementary Material where it is not strictly necessary to report them within the main body of the text. Please rethink the way in which you present the results and their discussion. Some more detailed comments are presented in the following:

Abstract

line 23-24: the statement about the scenarios and the non-reference one is not clear. Please explain better.

line 26-29: it is a bit of a mess; it could be summarized highlighting only the most important results.

Introduction

line 35-47: there are several concepts that are not developed at all. It could be more effective to explain in general the problem then report the current state of literature.

May I suggest under changing climate conditions, the effect of wildfires on landcover could play an important role as well. See, among others, Abbate, A., Longoni, L., Ivanov, V.I. and Papini, M., 2019. Wildfire impacts on slope stability triggering in mountain areas. Geosciences, 9(10), p.417.

line 48-69: well done description and very clear.

line 70-86: a better specific explanation of the Tetis model, Validation-Calibration procedure and the 25 climate scenarios should be required here. It seems to be quite general.

Materials and methods

line 94-95: the figure is clear but maybe a background map for the catchment can be useful as a spatial reference.

line 127-147: well done and clearly explained

line 149-150: add the reference to figure 1 for the location of rain gauges

line 178-196: well explained but a table with validation and calibration setting could be more effective

line 197-200: consider implementing graphics in pairs of 2 and try to enlarge a bit the legend and axes characters

line 215-231: Now climate scenarios is clearer because is explains what is intended, i.e. variation of precipitation, to move table 3 above that can follow directly the relative text.

line 232-255: Here there is too much information and it is not so clear. Please consider rewriting this paragraph in a more linear sequence:

                       1) grouping of the cathment 2) FDC explanation 3) Statistical Testing

                       Anyway, the approach is interesting

Results

line 258-274: again, that may be a table can summarize the data explained in the text

line 279-402: this way to present data is a bit disorienting. In my opinion, some tables can be avoided, and the graphs can be presented in a more clear way. First of all, the font of the plots should be readable, in particular figure 5 and 7, and then the data should be presented with a wiser logic. However, the elaboration is rather interesting.

line 403-520:  In my personal opinion, for this type of work result and discussion should be presented together. For the reader it would be simpler to follow. Try to compare graphic result and tables. Then the comments about the model limitation could be summarized a bit.

line 521-550: Conclusion are well done but could be better integrated with the result-discussion section, highlight some results obtained for example about the Pessimistic and Optimistic climate scenario and different groups of vegetation cover.

Author Response

REVIEWER 3

LINE 23-24: the statement about the scenarios and the non-reference one is not clear.

Please explain better.

Answer:

Reviewed, in section 2.3.4. the scenarios used are described

LINE 35-47: there are several concepts that are not developed at all. It could be more effective to explain in general the problem then report the current state of literature. May I suggest under changing climate conditions, the effect of wildfires on landcover could play an important role as well. See, among others, Abbate, A., Longoni, L., Ivanov, V.I. and Papini, M., 2019. Wildfire impacts on slope stability triggering in mountain areas. Geosciences, 9(10), p.417.

Answer:

Reviewed

LINE 70-86: a better specific explanation of the Tetis model, Validation-Calibration procedure and the 25 climate scenarios should be required here. It seems to be quite general.

Answer:

Reviewed

LINE 26-29: it is a bit of a mess; it could be summarized highlighting only the most important results.

Answer:

Reviewed

LINE 94-95: the figure is clear but maybe a background map for the catchment can be useful as a spatial reference.

Answer:

Reviewed

LINE 149-150: add the reference to figure 1 for the location of rain gauges

Answer:

Reviewed

LINE 178-196: well explained but a table with validation and calibration setting could be more effective

Answer:

Reviewed

LINE 197-200: consider implementing graphics in pairs of 2 and try to enlarge a bit the legend and axes characters

Answer:

Reviewed

LINE 215-231: Now climate scenarios is clearer because is explains what is intended, i.e. variation of precipitation, to move table 3 above that can follow directly the relative text.

Answer:

Reviewed

LINE 232-255: Here there is too much information and it is not so clear. Please consider rewriting this paragraph in a more linear sequence:

Answer:

Reviewed

LINE 258-274: again, that may be a table can summarize the data explained in the text

Answer:

Reviewed

LINE 279-402: this way to present data is a bit disorienting. In my opinion, some tables can be avoided, and the graphs can be presented in a more clear way. First of all, the font of the plots should be readable, in particular figure 5 and 7, and then the data should be presented with a wiser logic. However, the elaboration is rather interesting.

Answer:

Reviewed

LINE 403-520: In my personal opinion, for this type of work result and discussion should be presented together. For the reader it would be simpler to follow. Try to compare graphic result and tables. Then the comments about the model limitation could be summarized a bit.

Answer:

Reviewed

line 521-550: Conclusion are well done but could be better integrated with the result discussion section, highlight some results obtained for example about the Pessimistic and Optimistic climate scenario and different groups of vegetation cover.

Answer:

Reviewed

Round 2

Reviewer 1 Report

General comments

In relation to a followup review of this manuscript, I will limit myself to the main issues to decide whether I believe the manuscript has been significantly improved and now warrants publication in Water.   

  1. The authors have addressed my suggestions by making relatively minor changes to address a specific query I had in the annotated version of the initial manuscript. However, comments in which I suggested the authors provide more substantial additional information eg. Vegetation information or catchment climate information, has not been addressed adequately (ie. this information has not been provided). I also suggested that the authors provide more background as to why dry season flows were important (as these seemed to be an important driver to undertake this study). The authors claim to have provided additional information in the introduction, but I could find no evidence of this.

  1. The authors have responded to my question on how vegetation is represented in the model (but have not included this in the manuscript). It appears that vegetation was characterised using data from FAO Report 56. I don't think it is reasonable to conduct a study on the effects of vegetation cover on catchment hydrology, when there is no real test or validation of the vegetation (being simulated) transpiration and rainfall interception. If the approximate quantities of these processes is not understood, then one can not be certain of the resulting changes in streamflow when vegetation cover changes.

  1. My initial comment that only NSE is used to optimise and assess the models has been unanswered. I suggested also including values of mean and standard deviation of the model runs compared with the observed data to see whether the models have the total volumes and variation about right. The NSE values for both calibration and validation are so low that there is little confidence that the models are reproducing streamflow correctly.

  1. I asked the authors why they decided to express daily flows per second (m3/s)rather than per day and why they express flows as a volume rather than as a depth (to normalise for catchment area). No changes were made in the manuscript, and their response was that observed flow as input into the model exists as m3/s. I don't see why this can't be converted to daily estimates as is provided in other figures/tables in the manuscript. No appropriate reason was provided as to why the authors could not normalize for catchment area (as in other parts of the manuscript), other than they grouped the catchments by size. It is still not possible to compare the flow of catchments/groups of different sizes.

  1. There have been no major changes in the structure of the manuscript to reduce its length and reduce complexity of tables and figures to make it easier to follow. It remains very long, and difficult for the reader to understand how changes in climate and land cover interacted and what the effect in streamflow was from these simulations.

  1. The discussion claims this study inform forest management guidelines to minimise the impact of forest cover on streamflow, but this study only provides some modelling results, and did not discuss how this data could inform these guidelines. Are there existing guidelines? If, so, what do they recommend and how can this study support that?

Author Response

REVIEWER 1

General comments

In relation to a follow up review of this manuscript, I will limit myself to the main issues to decide whether I believe the manuscript has been significantly improved and now warrants publication in Water.

Answer:

First of all authors thank the reviewer for this new thorough revision, which again has help to improve the manuscript a lot. We have addressed the main issues raised by the reviewer, which we also hope are more clearly answered on this occasion. The paper is now much shorter and a significant proportion of the material has been moved to the Supplementary Material Section to enhance the readiness of the manuscript. We have revised and completed the list of references, providing an updated overview of the interactions between basin hydrology, land uses and climate. The next point-by-point try to provide answers to this revision satisfactorily and, in any case, we will be ready to undertake further improvements if required. 

- The authors have addressed my suggestions by making relatively minor changes to address a specific query I had in the annotated version of the initial manuscript. However, comments in which I suggested the authors provide more substantial additional information eg. Vegetation information or catchment climate information, has not been addressed adequately (ie. this information has not been provided). I also suggested that the authors provide more background as to why dry season flows were important (as these seemed to be an important driver to undertake this study). The authors claim to have provided additional information in the introduction, but I could find no evidence of this.

Answer:

1) Vegetation

Thanks for suggestion. Information on vegetation has been expanded in the text, lines 122 to 134 of the new version of the manuscript.

Vegetation covers from 56.3 to 92.7% for planted forests and from 4.0 to 40.9% in case of natural forests. Most of the area in the studied catchments is covered by Eucalyptus, whereas the steep hillslopes are occupied by Pinus radiata that have naturally invaded the riparian zone during forest rotations (Ulloa et al. 2011). Soils exhibit a thin herbaceous cover (<25%) mainly composed by grass that usually perishes during summer because of water deficit (Huber et al. 2010). The shrub cover is found under the most developed plantations and is rather disperse and mostly composed by genera Aristotelia and Rubus, as well as by some arboreal genera such as Luma, Peumus, Persea lingue and Nothofagus. Stream courses in all catchments are bounded by a strip of forest with an average width between 15 and 70 meters and composed mainly by native forest species of the genera Luma, Peumus, Persea lingue and Nothofagus. The remaining catchment area is dominated by forest plantations of the genera Eucalyptus and Pinus (Table 1).

2) Climate

Thanks for suggestion. Information on climate has been expanded in the text, lines 104 to 214 of the new version of the manuscript.

The study catchments are characterized by subtropical Mediterranean climate with dry summers. Mean long-term annual rainfall is estimated with 867 to 1421 mm, most of which (95 %) occurs between April and September during frequent and prolonged low- to moderate-intensity frontal storms. The long-term rainfall record is marked by inter-annual variations and its spatial distribution is topographically controlled by the nearby mountain range. During rainfall events, the plantations are generally immersed in mist or clouds due to the relatively low altitude of the catchments (233 to 389 m.a.s.l). On average, 25.5% (16.3 to 41.3%) of the annual rainfall turns into highly responsive runoff. The Mediterranean climate has wet, mild winters and exceptionally dry, cool summers. The temperature ranges from more than 40°C during summer to less than -3°C in winter, with an annual average of 13°C (Mohr et al. 2012).

3) Dry season

Thanks for suggestion. Information on dry season has been expanded in the text, lines 64 to 73 of the new version of the manuscript.

In addition, a decrease in the annual water yield, summer runoff and peak flows is associated with higher evapotranspiration from the forested parts of the catchments (mostly with Eucalyptus nitens and Pinus radiata (as previously reported elsewhere e.g.  mostly; species Bosch & Hewlett, 1982; Best, Zhang, McMahon, Western & Vertessy, 2003; Andréassian, 2004; Brown, Zhang, McMahon, Western & Vertessy, 2005). From the ecological and socio-economic points of views, expansion of planted forests has been questioned because of its impact on decrease streamflow, especially during the dry season (Iroumé & Palacios, 2013). Understanding the relationship between catchment runoff, forest management, and climatic factors, as well as their temporal evolution, is therefore crucial to developing integrated water management policies in forested catchments, especially in areas where water shortages are structural and frequent.

- The authors have responded to my question on how vegetation is represented in the model (but have not included this in the manuscript). It appears that vegetation was characterised using data from FAO Report 56. I don't think it is reasonable to conduct a study on the effects of vegetation cover on catchment hydrology, when there is no real test or validation of the vegetation (being simulated) transpiration and rainfall interception. If the approximate quantities of these processes is not understood, then one cannot be certain of the resulting changes in streamflow when vegetation cover changes. How was vegetation represented in the model? how did you parameterise each of the land use types win the model?

Answer:

This information is loaded into the model through two files related to the effect of evapotranspiration: i) the Land Cover Codes (cobveg) and the ii) Vegetation Factor or temporal distribution of the ET factor (FactorETmes.txt). The coverage index for Evapotranspiration modeling is calculated from the type of vegetation cover, based on a reclassification of the vegetation / land use map. For this, each cell is assigned a numerical value corresponding to the vegetation type. An annual variation of evaporation is assigned to this numerical value in the vegetation factor table (ETmes). The vegetation factor (FAO crop factor) represents the variability of evaporation in the annual cycle. It indicates the growth status or the amount of biomass of each type of vegetation. For each category defined in relation to the vegetation cover, an evapotranspiration factor value is assigned. The values ​​have been obtained from FAO Report 56: "Crop evaporation: Guides for determining crop water requirements" (fao.org). In turn, the ETmes factor file has 14 columns: 12 correspond to the evapotranspiration values ​​of each month, the penultimate is the interception value of the vegetation, which is also included in this file (later the interception is explained), and the last corresponds to the vegetation codes used. Columns always separated by spaces. This file does not have headers of any kind, for example:

* Example

1.00 1.00 1.00 1.00 1.00 1.00 1.00 1.00 1.00 1.00 1.00 1.00 9 01- Evergreen Forest

0.60 0.60 0.80 1.00 1.00 1.00 1.00 1.00 1.00 1.00 0.80 0.60 9 02- Deciduous forest

0.60 0.60 0.60 0.60 0.60 0.60 0.60 0.60 0.60 0.60 0.60 0.60 1 03- Perennial arboreal crop

 - My initial comment that only NSE is used to optimise and assess the models has been unanswered. I suggested also including values of mean and standard deviation of the model runs compared with the observed data to see whether the models have the total volumes and variation about right. The NSE values for both calibration and validation are so low that there is little confidence that the models are reproducing streamflow correctly. My initial comment that only NSE is used to optimise and assess the models has been unanswered = Please include information on at what time step the model was run and at what time step NSE was calculated. eg. model was run on a daily time step and NSE calculated monthly

Answer:

The precision in the simulation or prediction is given by the measurement of the differences between what is simulated or predicted and what is actually observed. These differences can be systematic (recurring) or random (Lettenmaier and Wood, 1992). There are numerous criteria to define the objective function within the automatic optimization process. The TETIS model uses the following criteria:

  • Volume error
  • Root mean square error RMSE
  • efficiency coefficient NSE
  • Kling and Gupta Index
  • Log Bias NSE
  • The error of the logarithms
  • Autocorrelated gaussian error
  • Hererocedastic maximum likelihood estimator HMLE

We agree with the reviewer’s point of view about the interest of different indicators to evaluate the quality of the obtained results; and within them it is worth noting that NSE is the most commonly used for model evaluation, since it involves standardization of the residual variance and its expected value does not change with the length of the record or the magnitude of the runoff (Kachroo and Natale, 1992; cited in Kothyari and Singh, 1999). There were also several reasons why we used NSE to compare the different results. The first is that it is a universal indicator especially suitable for comparing temporal data series in hydrological modelling, as it is highly sensible to the proper reproduction of extremes and irregularities in the data series. Secondly, the model we used to simulate all the catchments and scenarios provides NSE in a quite straightforward manner, so taking into account the high number of simulations involved in the study and the computational time associated with each of them, NSE was the best candidate to have a way to compare the results from the whole set of simulations in a reasonable time. We agree that many of the values of NSE are low, but nevertheless we think that the obtained results allow the comparison between the different situations and provide useful conclusions about the observed trends. Overall, in this study, NSE was evaluated on a daily time step and end of each data series and if possible, we would like to keep it like this, based in the following rating criteria (Moriasi et al., 2015): not satisfactory (NSE ≤ 0.50), satisfactory (0.50 < NSE ≤ 0.70), good (0.70 < NSE ≤ 0.80) and very good (NSE > 0.80). On the other hand, values of mean and standard deviation of the model runs, are detailed in table S8.

- I suggested also including values of mean and standard deviation of the model runs compared with the observed data to see whether the models have the total volumes and variation about right

Answer:

Additional information of Table 4. Can be found online in the Supporting Information section at the end of the article (Table S8). In Table S4 there we include values of mean and standard deviation of the model runs compared with the observed data to see whether the model performed satisfactory with compare observed data

- I asked the authors why they decided to express daily flows per second (m3/s) rather than per day and why they express flows as a volume rather than as a depth (to normalise for catchment area). No changes were made in the manuscript, and their response was that observed flow as input into the model exists as m3/s. I don't see why this can't be converted to daily estimates as is provided in other figures/tables in the manuscript. No appropriate reason was provided as to why the authors could not normalize for catchment area (as in other parts of the manuscript), other than they grouped the catchments by size. It is still not possible to compare the flow of catchments/groups of different sizes.

Why have you decided to express daily flows per second rather than per day? Also why have you decided to express flows as a volume rather than as a depth (to normalise for catchment area)?

Answer:

The model requires and delivers a database in m3/s. It was decided to work with the same hydrograph output format of Figure S1 for the rest of the figures. Indeed, values can be expressed in (l/s) or (mm), but this apparently simple change would require new analysis, with lots of changes in the text, figures and tables. We would like to remark that there is no inter-basin comparison whatsoever in the paper, therefore there is no need to undertake such analysis. The influence of land use and climate change under different is done within each catchment individually, therefore there is no fundamental need to normalize the values, so not to change the logic of the document. Worth to notice that, occasionally, if a comparison is made (i.e. rainfall), standardized values are used (see next answer).

- It is not surprising that there are differences based on catchment size when you have compared flow volumes that do not normalise for catchment size.

Answer:

Lines 277 to 294 of the new version of the manuscript: MW and KW test methods were performed to compare between rainfall stations and between catchments (with values ​​in mm).

Lines 312 to 315 of the new version of the manuscript: MW and KW tests were used to compare in each catchment the observed vs the simulated flow (​​in m3/s). As comparisons between catchments were not performed, we consider it was not necessary to standardize by the area of ​​each catchment.

Lines 344 to 365 of the new version of the manuscript: Comparison made between observed and simulated flows under three land use scenarios per catchment. As comparisons between catchments were not performed, we consider no standardization was required.

Lines 366 to 368 of the new version of the manuscript: Statistical comparison between observed and simulated flows by catchment. The flows Q5, Q16, Q50, Q84 and Q95 were obtained using flow duration curves (FDC) from daily data for each of the catchments to characterize the temporal distribution of the discharges and detect changes within the various simulations. From FDCs we obtained percentile values of Q5, Q16, Q50, Q84 and Q95; where Qi is the value corresponding to ith percentile, i.e., the discharges equal to or exceeding 5, 16, 50, 84, and 95% of the time, respectively; therefore, the temporal distribution of the flows within each catchment is compared.

Line 369 to 372 of the new version of the manuscript: Comparison using the temporal distribution (FDC); the distribution of the flows is compared between catchments, emphasizing in the text the flows Q5, Q16, Q50, Q84 and Q95. This paragraph was eliminated.

Lines 324 to 337 of the new version of the manuscript: Comparison of the total volume observed and the total volume simulated under different land use scenarios.

- There have been no major changes in the structure of the manuscript to reduce its length and reduce complexity of tables and figures to make it easier to follow. It remains very long, and difficult for the reader to understand how changes in climate and land cover interacted and what the effect in streamflow was from these simulations.

Answer:

The reviewer is right and his/her recommendation has been taken into account. The paper has been reviewed to reduce its length. For this, several figures and tables are now additional supporting materials, which can now be found in the online Supporting Information section at the end of the paper.

- The discussion claims this study inform forest management guidelines to minimise the impact of forest cover on streamflow, but this study only provides some modelling results, and did not discuss how this data could inform these guidelines. Are there existing guidelines? If, so, what do they recommend and how can this study support that?

Answer:

This has been reviewed and changed. We sincerely believe that the study sheds light on the effects of land management on streamflows and the information is relevant to support discussions related to forest management, intended to minimize the impact of forest cover on water yield, altogether in a climate change scenario, whose effects are expected to be acute in countries like Chile. Specifically, the work focuses on small catchments as they are likely to be most affected by the combination of land-use and climate changes.

Reviewer 2 Report

Dear Authors,

Dear Editor,

Definitely, you improved the manuscript after the first round of major revision. 

But I still think it is too long! It is difficult to follow.

For instance, I am geomorphologist (fluvial geoorphologist) with background in forestry domain. Apparently, this manuscript should be easy to follow for me, but it is not....

But again, I am still impressed about the amount of data and work presented.

I red again and again the manuscript, I do not know what to say!!! For e.g.  the graphical illustration...it is impressive, clear, but still hard to follow... 

I put myself in the authors' shoes......I know, it is very hard to cut and synthesize, but I would try to do this, in the end.

I really do not know what decision to make...

I do not dispute the scientific quality, but I think of those who will read and cite this article in the future. For me, until now, this is the longest manuscript to read and review.

Dear Editor, the decision, is yours. I hope the other 2 reviewers will help you more.

This paper is valuable; is supported by a lot of data; high-resolution analysis and data; a lot of GIS; statistics; modelling....(Maybe you could write at least 2-3 papers by better synthesizing this valuable data).

The manuscript .pdf is now 70 pages...

I will agree to a major revision with 2 suggestions/recommendations:

1. You will rewrite the manuscript by synthesizing better and resend the manuscript

2. If other reviewers and the Editor will accept the manuscript, in the present form! (I agree with them, also).

All the best,

Author Response

REVIEWER 2

Definitely, you improved the manuscript after the first round of major revision. But I still think it is too long! It is difficult to follow. For instance, I am geomorphologist (fluvial geoorphologist) with background in forestry domain. Apparently, this manuscript should be easy to follow for me, but it is not.... But again, I am still impressed about the amount of data and work presented. I red again and again the manuscript, I do not know what to say!!! For e.g.  the graphical illustration...it is impressive, clear, but still hard to follow...  I put myself in the authors' shoes......I know, it is very hard to cut and synthesize, but I would try to do this, in the end. I really do not know what decision to make... I do not dispute the scientific quality, but I think of those who will read and cite this article in the future. For me, until now, this is the longest manuscript to read and review. Dear Editor, the decision, is yours. I hope the other 2 reviewers will help you more. This paper is valuable; is supported by a lot of data; high-resolution analysis and data; a lot of GIS; statistics; modelling....(Maybe you could write at least 2-3 papers by better synthesizing this valuable data). The manuscript .pdf is now 70 pages... I will agree to a major revision with 2 suggestions/recommendations:

  1. You will rewrite the manuscript by synthesizing better and resend the manuscript
  2. If other reviewers and the Editor will accept the manuscript, in the present form! (I agree with them, also).

Answer:

We again thank the reviewer for his/her very positive comments. We agree that the manuscript was unnecessarily long and was not easy to follow. We reduced its length to make it clearer. The manuscript has now 22 pages, including references. In addition, several tables and figures are now included as additional supporting materials, which can be found online in the Supporting Information section at the end of the paper. We hope that by doing this, the paper flows now more clearly and it is easier to read. 

Reviewer 3 Report

The structure of the revised manuscript is much clearer and better organized with respect to the original version. However, I am not satisfied with authors' responses in what replying with 'Reviewed' is not acceptable for the standards of the journal. Since the manunscript has undergone some subsantial revision, the corrections should be indicated with their line number corresponding to the new manuscript and explained in the cover letter accordingly. I would like to add that the graphical presentation of the figures must be further improved in order to increase readability to an acceptable level. In particular: 

  • Line 212: could you display figures in pairs, specify the letter a) b) etc… with the corresponding name of the catchment, enlarging the axes scale and font characters?
  • Line 227: here you used N2,N3 etc… instead above is a) b) c)… Please use one unique notation es. N2,N3.. also for the above graphs.
  • Line 290: Enlarge legend characters and in the first graph the different catchments are not distinguishable, especially if the paper is printed in b-w.
  • Line 312: Please Resize the Graphs, Legends are too little and add a more complete caption explaining the difference among Whole Simulation and Drier Season Calibrations.
  • Line 378: the same as 312.
  • Line 394: the same as 332. Too much information are not readable also zooming. Please consider another way to present them.

You have explained in the introduction an also remarked in the conclusions the important impact of land-use in catchment discharge modelling. But have you considered the climate change effects due to the increase in the duration of drier seasons? Forest or other type of cultivations may incur wildfires and this can have dramatic effects on basin water discharge, triggering diffused hydrogeological issues. This is a real problem that is hugely studied in Mediterranean catchments, that are now starting to experience these effects. In this light I strongly suggest to consider that under changing climate conditions, the effect of wildfires on landcover could play an important role as well. See, among others, Abbate, A., Longoni, L., Ivanov, V.I. and Papini, M., 2019. Wildfire impacts on slope stability triggering in mountain areas. Geosciences, 9(10), p.417.

Author Response

REVIEWER 3

The structure of the revised manuscript is much clearer and better organized with respect to the original version. However, I am not satisfied with authors' responses in what replying with 'Reviewed' is not acceptable for the standards of the journal. Since the manuscript has undergone some substantial revision, the corrections should be indicated with their line number corresponding to the new manuscript and explained in the cover letter accordingly. I would like to add that the graphical presentation of the figures must be further improved in order to increase readability to an acceptable level.

- LINE 23-24: the statement about the scenarios and the non-reference one is not clear. Please explain better.

Answer:

This has been reviewed in section 2.3.4. The explanation on the three land cover scenarios information can be found online in the Supporting Information section at the end of the article (Figure S7).

- LINE 35-47: there are several concepts that are not developed at all. It could be more effective to explain in general the problem then report the current state of literature. May I suggest under changing climate conditions, the effect of wildfires on landcover could play an important role as well. See, among others, Abbate, A., Longoni, L., Ivanov, V.I. and Papini, M., 2019. Wildfire impacts on slope stability triggering in mountain areas. Geosciences, 9(10), p.417.

Answer:

This comment is really appropriate given the nature of the study area; despite this, this issue is not in the scope of the paper and is important to note that during the study period there were no forest fires, therefore, their potential effects were not considered in the model.

- LINE 70-86: a better specific explanation of the Tetis model, Validation-Calibration procedure and the 25 climate scenarios should be required here. It seems to be quite general.

Answer:

This was reviewed. The three land cover scenarios information can be found online in the Supporting Information section at the end of the article (Figure S2). Climate scenarios information are online in the Supporting Information section at the end of the article (Table S1).

- LINE 26-29: it is a bit of a mess; it could be summarized highlighting only the most important results.

Answer:

This issue has been reviewed as suggested by the reviewer. Results are presented by catchments size i.e. called small and large owing to the scale of the study, suggesting that future changes in water yield may be influenced to a greater extent by a 20% decrease in precipitation and a 20% increase in evapotranspiration. These future changes in water yield will occur to a greater extent in smaller catchments (Lines 26 to 32 of the new version of the manuscript).

- LINE 94-95: the figure is clear but maybe a background map for the catchment can be useful as a spatial reference.

Answer:

This has been reviewed and the spatial location is indicated in red circles (Figure 1).

- LINE 149-150: add the reference to figure 1 for the location of rain gauges

Answer:

This has been reviewed and the location of the rain gauges added to Figure 1.

- LINE 178-196: well explained but a table with validation and calibration setting could be more effective

Answer:

This has been reviewed and the Table 3 informs of the NSE values for calibration and validation. Additional information of Table 3 can be found online in the supporting information section at the end of the article (Figure S1).

- LINE 197-200: consider implementing graphics in pairs of 2 and try to enlarge a bit the legend and axes characters

Answer:

This has been reviewed and changed. Legend and axis characters are new in all the figures.

- LINE 215-231: Now climate scenarios is clearer because is explains what is intended, i.e. variation of precipitation, to move table 3 above that can follow directly the relative text.

Answer:

Thanks for this comment. As indicated, this has been reviewed and the Table 3 informs of the NSE values for calibration and validation. Additional information of Table 3 can be found online in the supporting information section at the end of the article (Figure S1).

- LINE 232-255: Here there is too much information and it is not so clear. Please consider rewriting this paragraph in a more linear sequence:

Answer:

This has been reviewed. Additional information can be found online in the supporting information section at the end of the article (Table S2).

- LINE 258-274: again, that may be a table can summarize the data explained in the text

Answer:

This has been reviewed. Additional information of Figure 2b can be found online in the supporting information section at the end of the article (Table S5)

LINE 279-402: this way to present data is a bit disorienting. In my opinion, some tables can be avoided, and the graphs can be presented in a more clear way. First of all, the font of the plots should be readable, in particular figure 5 and 7, and then the data should be presented with a wiser logic. However, the elaboration is rather interesting.

Answer:

This has been reviewed and legends and axis characters changed in all figures. The number of figures in the main text has been substantially reduced, and other figures are now presented as additional supporting information.

- LINE 403-520: In my personal opinion, for this type of work result and discussion should be presented together. For the reader it would be simpler to follow. Try to compare graphic result and tables. Then the comments about the model limitation could be summarized a bit.

Answer:

We sincerely thank the reviewer comment. This is always a controversial issue with reviewers having contrasting opinions. We accept that he/she might be right but this would imply a substantial re-writing of the manuscript. If possible we would like to keep it like this. None of the other reviewers have recommended this.

- LINE 521-550: Conclusion are well done but could be better integrated with the result discussion section, highlight some results obtained for example about the Pessimistic and Optimistic climate scenario and different groups of vegetation cover.

Answer:

Again, we sincerely thank the reviewer comment. As for the results and discussion, including the conclusions within the discussion is always a possibility. In this case we would like to keep a conclusions section (we call it ‘Final remarks’ since we thought that ‘Conclusion’ was a too strong word for a modelling study such as this one). In this final section we only try to state the main findings (or take-home messages) of the paper, and send a concise final message to the reader, being aware that results of a modelling exercise cannot be totally conclusive. If possible we would like to keep it like this; but if it turns mandatory we can integrate the conclusions in the discussion section since this change does not imply a major restructuring of the paper as the previous one. None of the other reviewers have recommended this either.

- In particular:

- LINE 212: could you display figures in pairs, specify the letter a) b) etc… with the corresponding name of the catchment, enlarging the axes scale and font characters?

This has been reviewed and legends and axis characters changed in all figures. The number of figures have been substantially reduced and are now presented as additional supporting information.

- LINE 227: here you used N2,N3 etc… instead above is a) b) c)… Please use one unique notation es. N2,N3.. also for the above graphs.

Again, this has been reviewed and legends and axis characters changed in all figures. The number of figures have been substantially reduced and are now presented as additional supporting information.

- LINE 290: Enlarge legend characters and in the first graph the different catchments are not distinguishable, especially if the paper is printed in b-w.

Indeed, this has been reviewed and legends and axis characters changed in all figures. The number of figures have been substantially reduced and are now presented as additional supporting information.

- LINE 312: Please Resize the Graphs, Legends are too little and add a more complete caption explaining the difference among Whole Simulation and Drier Season Calibrations.

Yes, this has been reviewed and legends and axis characters changed in all figures. The number of figures have been substantially reduced and are now presented as additional supporting information.

- LINE 378: the same as 312.

This has been reviewed and legends and axis characters changed in all figures. The number of figures have been substantially reduced and are now presented as additional supporting information.

- LINE 394: the same as 332. Too much information are not readable also zooming. Please consider another way to present them.

This has been reviewed and legends and axis characters changed in all figures. The number of figures have been substantially reduced and are now presented as additional supporting information.

- You have explained in the introduction an also remarked in the conclusions the important impact of land-use in catchment discharge modelling. But have you considered the climate change effects due to the increase in the duration of drier seasons? Forest or other type of cultivations may incur wildfires and this can have dramatic effects on basin water discharge, triggering diffused hydrogeological issues. This is a real problem that is hugely studied in Mediterranean catchments, that are now starting to experience these effects. In this light I strongly suggest to consider that under changing climate conditions, the effect of wildfires on landcover could play an important role as well. See, among others, Abbate, A., Longoni, L., Ivanov, V.I. and Papini, M., 2019. Wildfire impacts on slope stability triggering in mountain areas. Geosciences, 9(10), p.417.

Answer

As stated, this comment is really appropriate given the nature of the study area; despite this, this issue was never in the scope of the paper, and it is important to note that during the study period there were no forest fires, therefore, their potential effects were not considered in the model.

Round 3

Reviewer 1 Report

General comments

While I had my reservations whether this manuscript would ever undergo the required changes in order to make it publishable, I do commend the authors on their persistence in attempting to do that. They have made serious attempts to address most of my earlier suggestions eg. length of manuscript.

I will limit my third review to the main issues that still remain, and will leave the final decision to the editors as to whether it is published in its present for, but in my, view further revision is required.

1. The authors have now included details on how vegetation is represented in the model. Vegetation is represented in the model by an ET factor for each vegetation type. Actual water use by each vegetation type is then calculated using this factor and the ET input into the model. While the mean annual rainfall is sufficient supply of water for forests, most of this ~95% falls between April and September, and summers are dry and are likely to result in limited water availability to forests in the region. The ET factor used (1.0 throughout the year) for forests does not appear take this into account. It seems that the authors have taken the generic FAO Report 56 factors, and applied them to forests in Chile. It is mentioned elsewhere that measured temperature was used by the model to calculate evapotranspiration. At no stage is there any validation on whether the estimated water use (transpiration or rainfall interception) of the two main land uses (forests and pasture) even approximate actual measured estimates. Therefore, my earlier concerns still stand - that I don't think it is reasonable to conduct a study on the effects of vegetation cover on catchment hydrology, when there is no real test or validation of the vegetation (being simulated) transpiration and rainfall interception. If the approximate quantities of these processes are not understood, then one cannot be certain of the resulting changes in streamflow when vegetation cover changes.

2. I previously asked the authors why they decided to express daily flows per second (m3/s) rather than per day and why they express flows as a volume rather than as a depth (to normalise for catchment area). Their response continues to be along the lines that the model they use requires and delivers a database in m3/s and therefore they decided to use the same units. They also claim it would require a substantial amount of work and re-analysis. However, Figure 2 provides runoff data in mm/d, while others do not. Table S3 provides runoff data in mm/d, yet Tables S4 and S5 (and others) provides it in m3 over time – so there is some confusion there. Not only does quoting flows in volume rather than as depth normalised for catchment area make it difficult to compare between catchments (comparison which the authors claim they do not do) but it makes it more difficult for other researchers to draw comparisons without first having to calculate normalised flows.

Author Response

REVIEWER 1

  • While I had my reservations whether this manuscript would ever undergo the required changes in order to make it publishable, I do commend the authors on their persistence in attempting to do that. They have made serious attempts to address most of my earlier suggestions eg. length of manuscript. I will limit my third review to the main issues that still remain, and will leave the final decision to the editors as to whether it is published in its present for, but in my, view further revision is required.

Answer:

First of all authors thank the reviewer for this new thorough revision, which again has help to improve the manuscript a lot. We have addressed the main issues raised by the reviewer, which we also hope are more clearly answered on this occasion.

  • The authors have now included details on how vegetation is represented in the model. Vegetation is represented in the model by an ET factor for each vegetation type. Actual water use by each vegetation type is then calculated using this factor and the ET input into the model. While the mean annual rainfall is sufficient supply of water for forests, most of this ~95% falls between April and September, and summers are dry and are likely to result in limited water availability to forests in the region. The ET factor used (1.0 throughout the year) for forests does not appear take this into account. It seems that the authors have taken the generic FAO Report 56 factors, and applied them to forests in Chile. It is mentioned elsewhere that measured temperature was used by the model to calculate evapotranspiration. At no stage is there any validation on whether the estimated water use (transpiration or rainfall interception) of the two main land uses (forests and pasture) even approximate actual measured estimates. Therefore, my earlier concerns still stand - that I don't think it is reasonable to conduct a study on the effects of vegetation cover on catchment hydrology, when there is no real test or validation of the vegetation (being simulated) transpiration and rainfall interception. If the approximate quantities of these processes are not understood, then one cannot be certain of the resulting changes in streamflow when vegetation cover changes.

Answer:

"We understand the point of view of the reviewer and we agree that it would be optimal to be able to calibrate independently interception and evapotranspiration processes. Unfortunately, no suitable data are available to carry out this calibration. Despite this inconvenient, we think that the way we proceeded is consistent enough to perform the analysis we are dealing with. The ETmes factors reflect changes in vegetation characteristics throughout the year whereas meteorological data reflect the irregularity of precipitation. Two of the calibration factors involved in the model are directly linked to static storage (FC1) and evapotranspiration (FC2). Therefore, when we calibrate the model with the available discharge data, we are also calibrating this two coefficients, even if this calibration is performed within the whole set of calibration parameters. The irregularity in precipitation is not considered in FactorETMes, but is considered in precipitation input data and propagated through the model by means of the mass balance equations that describe the different processes. As a conclusion, despite some limitations, we think that the approach is valid to provide a basis for the analysis performed in this article."

  • I previously asked the authors why they decided to express daily flows per second (m3/s) rather than per day and why they express flows as a volume rather than as a depth (to normalise for catchment area). Their response continues to be along the lines that the model they use requires and delivers a database in m3/s and therefore they decided to use the same units. They also claim it would require a substantial amount of work and re-analysis. However, Figure 2 provides runoff data in mm/d, while others do not. Table S3 provides runoff data in mm/d, yet Tables S4 and S5 (and others) provides it in m3 over time – so there is some confusion there. Not only does quoting flows in volume rather than as depth normalised for catchment area make it difficult to compare between catchments (comparison which the authors claim they do not do) but it makes it more difficult for other researchers to draw comparisons without first having to calculate normalised flows.

Answer:

Thanks for suggestion. The figures and tables were changed to mm, as suggested.

Line 317 of the new version of the manuscript: Figure 3.

Line 390 of the new version of the manuscript: Figure 4.

Line 400 of the new version of the manuscript: Figure 5

Line 338 of the new version of the manuscript: Additional supporting information Table S4

Line 343 of the new version of the manuscript: Additional supporting information Table S5

Reviewer 2 Report

Dear authors,

I still consider that this version of the new delivered manuscript is too long!

There are so many data and unsinthetized materials and ideas, that until at the end of manuscript, it is hard to follow!

But, I can notice some changes, good changes since the last version of the manuscript!

It might be considered for publication, if the editor have good feeback from all of us and him/herself!

I maintain my opinion, that we deal with good data here, that can be usefull for the other researchers.

Specific comments:

Where is 4.2 from Discussion section?

As we can observe, your proposed model has a lot of limitations. By taking this aspect into consideration, do you think that this model can be extrapolate to other climatic regions? can we ”force” to use this model? Can you make some recommendations for others trying to extrapolate it in order to simulate or elaborate future projections? 

All the best

Author Response

REVIEWER 2

  • I still consider that this version of the new delivered manuscript is too long! There are so many data and unsinthetized materials and ideas, that until at the end of manuscript, it is hard to follow! But, I can notice some changes, good changes since the last version of the manuscript! It might be considered for publication, if the editor have good feeback from all of us and him/herself! I maintain my opinion, that we deal with good data here, that can be usefull for the other researchers.

Answer:

First of all, authors thank the reviewer for this new thorough revision, which again has help to improve the manuscript a lot. We have addressed the main issues raised by the reviewer, which we also hope are more clearly answered on this occasion. We agree that the manuscript was unnecessarily long and was not easy to follow. We have again reduced its length to make it clearer. The manuscript has now 22 pages, including references, which is an average length for most manuscripts. In addition, several tables and figures are now included as additional supporting materials, which can be found online in the Supporting Information section at the end of the paper. We hope that by doing this, the paper flows now more clearly and it is easier to read. 

  • Where is 4.2 from Discussion section?

Answer:

Reviewed and changed: Line 475 of the new version of the manuscript: Model Limitations

  • As we can observe, your proposed model has a lot of limitations. By taking this aspect into consideration, do you think that this model can be extrapolate to other climatic regions? can we ”force” to use this model? Can you make some recommendations for others trying to extrapolate it in order to simulate or elaborate future projections? 

Answer:

Thanks for the suggestion. Hydrological modeling has traditionally been a tool to understand the processes of the hydrological cycle. Therefore, the reliable estimation of the state variables involved ultimately depends on the conceptual characteristics of the selected model. Today, distributed modeling is one of the key tools for the estimation and prediction of flood events and for the evaluation of water resources. The advantages of distributed modeling and especially the TETIS model with respect to traditional aggregate modeling consist fundamentally in the best representation of the spatial variability of the phenomena involved in hydrological processes; although sometimes this requires detailed information on the study area, often not available or scarce.

Model performance emphasizes its potential as a tool for evaluating water and sediment yield for large catchments, as well as of its usefulness for water and sediment management in light of future climate and land use change scenarios. This model can be extrapolate to other climatic regions, as it is documented in a series of scientific articles; http://lluvia.dihma.upv.es/es/publi/artic.html

Future work can be directed in two ways. On the one hand, to generate higher quality and the amount of data available, for example, with respect to the physical variables of the soil, and secondly, in the simplification of the information, either for the use of TETIS or another program that requires less detail. It will also be necessary to scale to larger catchments, in order to better validate the TETIS Model on a wider variety of forested catchments in Chile.

Reviewer 3 Report

The manuscript has been considerably improved. The decision to take much of the plots and tables to a supplementary file makes it easier to follow. One minor issues is, in my opinion, the quality of the figures. It is often hard to read the axes and zooming in does not help as the text becomes pixelated. I strongly suggest you to produce the graphs with a higher quality. 

Author Response

REVIEWER 3

  • The manuscript has been considerably improved. The decision to take much of the plots and tables to a supplementary file makes it easier to follow. One minor issues is, in my opinion, the quality of the figures. It is often hard to read the axes and zooming in does not help as the text becomes pixelated. I strongly suggest you to produce the graphs with a higher quality. 

Answer:

Thanks for suggestion. The quality of the figures was changed and uploading in pdf format.
